# A taxon-specific measurement of disruption in a multi-modal study of microbiomes and metabolomes reveals system-wide dysbiosis preceding HIV-1 infection

F. Fouladi[1], Y. Chen [2], S. Bera[1], A. K. Jarmusch [3], D. Van Tyne [2], F. J. Palella[4], J. B. Margolick[5], K. W. Chew [6], J. Sun [7], J. Martinson [8], C. R. Rinaldo[2] & S. D. Peddada [1] ✉

The microbiome plays an important role in immune responses and inflammation in HIV-1 infection. Hence, a deeper understanding of the changes in the microbiome, its function and metabolites, and their interactions prior to HIV-1 infection is potentially important for HIV-1 prevention strategies. Using stool, oral washes, and plasma biospecimens obtained from men who have sex with men (MSM) and who were without HIV-1, we found several differences in microbial ecologies, gene functions, and metabolites between MSM who became HIV-1 infected (Pre-HIV) within six months and those who remained HIV-1 uninfected (Non-HIV). The Pre-HIV group had an enrichment of enzymes involved in purine metabolism, lower amino acid metabolism, and higher oxidative stress before the infection compared to the Non-HIV group. We also introduced a novel and broadly applicable taxon-specific measure of DISruption in COrrelations (DISCO) with other features, such as microbial taxa and metabolites in a given group (e.g., Pre-HIV group) relative to a reference group (e.g., Non-HIV group). Using DISCO, we identified several gut and oral species with disrupted correlations prior to HIV-1 infection. Application of DISCO to external datasets revealed that *Prevotella spp.* are consistently disrupted in their correlations across multiple cohorts prior to or following HIV-1 infection.

The intestinal mucosa harbors the majority of CD4[+] T cells expressing C-C chemokine receptor type 5 (CCR5), the principle coreceptor for human immunodeficiency virus type 1 (HIV-1), and thereby, is the major site of HIV-1 entry and replication[1]. Given the intricate interaction between the gut microbiome and mucosal immune cells, primary HIV-1 infection and the subsequent mucosal inflammation are associated with disruption in the gut microbiome, known as "gut dysbiosis". Numerous studies have shown that people with HIV-1 infection (PWH) have a different microbial composition compared to people without HIV-1 infection (PWOH), with depletion of commensal short-chain fatty acid-producing bacteria, such as *Bifidobacterium, Bacteroides, Alistipes, Akkermansia,* and *Blautia*[2–4]. In contrast, abundances of potentially pathogenic and proinflammatory bacteria, such as *Prevotella, Catenibacterium, Bulleidia, Dialister, Mitsuokella,* and *Desulfovibrio* increase following HIV-1 infection[4,5].

While it is well established that gut dysbiosis occurs after the onset of HIV-1 infection[2,4–6], and that a potential causal link exists between the gut microbiome alterations and immune and metabolic dysregulation post-infection[7], emerging literature suggests that changes in the gut microbial composition and increases in proinflammatory

**Table 1 | Demographic characteristics of participants from the Multicenter AIDS Cohort Study (MACS)**

| Characteristic | Non-HIV N = 152[a] | Pre-HIV N = 87[a] | p-value[b] |
|---|---|---|---|
| Age | 42.5 (16.6) | 37.6 (14.7) | 0.029 |
| Location | | | 0.2 |
| Baltimore | 47.0 (30.9%) | 16.0 (18.4%) | |
| Chicago | 38.0 (25.0%) | 26.0 (29.9%) | |
| Los Angeles | 41.0 (27.0%) | 28.0 (32.2%) | |
| Pittsburgh | 26.0 (17.1%) | 17.0 (19.5%) | |
| Race | | | 0.11 |
| Black | 3.0 (2.0%) | 5.0 (5.7%) | |
| Other | 4.0 (2.6%) | 0.0 (0.0%) | |
| White | 144.0 (95.4%) | 82.0 (94.3%) | |
| Unknown | 1 | 0 | |
| Education | | | 0.13 |
| No degree | 50.0 (33.1%) | 37.0 (43.0%) | |
| Postgrad | 58.0 (38.4%) | 34.0 (39.5%) | |
| Undergrad | 43.0 (28.5%) | 15.0 (17.4%) | |
| Unknown | 1 | 1 | |
| Drinking | | | 0.075 |
| Heavy drinking | 75.0 (50.3%) | 53.0 (63.1%) | |
| Not heavy drinking | 74.0 (49.7%) | 31.0 (36.9%) | |
| Unknown | 3 | 3 | |
| Smoking | | | 0.11 |
| Current | 56.0 (37.8%) | 44.0 (51.8%) | |
| Former | 29.0 (19.6%) | 14.0 (16.5%) | |
| Never | 63.0 (42.6%) | 27.0 (31.8%) | |
| Unknown | 4 | 2 | |
| Antibiotic use | 61.0 (40.1%) | 53.0 (60.9%) | 0.003 |
| Any substance use | 112.0 (75.2%) | 78.0 (91.8%) | 0.002 |
| Unknown | 3 | 2 | |
| Number of receptive anal inter-course partners | | | <0.001 |
| 0 | 59.0 (39.3%) | 3.0 (3.4%) | |
| 1 | 40.0 (26.7%) | 17.0 (19.5%) | |
| 2–5 | 43.0 (28.7%) | 41.0 (47.1%) | |
| 6 or more | 8.0 (5.3%) | 26.0 (29.9%) | |
| Unknown | 2 | 0 | |

[a]Mean (SD); n (%).
[b]Two-sided Wilcoxon rank sum test; Fisher's exact test.

immune biomarker levels occur in men who have sex with men (MSM) even before the onset of HIV-1 infection[3,8–10]. Moreover, Lin et al.[9] found that several commensal and proinflammatory bacterial species, together with immune biomarkers such as sCD14 and sCD163, mediate the effect of sexual activity on the onset of HIV-1 infection. These findings suggest that the dysbiosis of the gut microbiome, rather than being merely a consequence of post-infection inflammation and immune response, could increase the risk of HIV-1 infection.

Since there is currently no cure or vaccine for HIV-1 infection, and the global incidence of HIV-1 infection has remained high, with an estimated 1.3 million new infections in 2024[11,12], modulation of the microbiome could hold promise for the control or prevention of HIV-1 infection. Towards this end, using the same study population in refs. 8,9, we performed a comprehensive analysis of the oral and gut microbiomes as well as short-chain fatty acids (SCFA) and untargeted metabolites within these sites, along with those circulating in plasma, in MSM enrolled in the Multicenter AIDS Cohort Study (MACS[13]; now part of the MACS-WIHS Combined Cohort Study[14]). Our system-level

analysis not only strengthens recent findings regarding the differences in gut ecology prior to the onset of HIV-1 infection but also reveals differences in gene ontology (GO) terms, referred as gene functions in this paper, as well as microbial products. We also introduce a novel taxon-specific measure of disruption in correlations called DISCO (DISruption in COrrelations) that identifies gut and oral species that were disrupted in their interactions with other species, their functions, and metabolites before the onset of HIV-1 infection.

## Results

This study includes 239 MSM enrolled in the MACS between 1984 and 1985[13]. While all 239 participants were without HIV-1 during their first MACS clinic visit, 87 were infected with HIV-1 by their second clinic visit approximately 6 months later (termed Pre-HIV) and the other 152 participants remained HIV-1 negative (termed Non-HIV). In fact, of these 152 Non-HIV, only two became positive for HIV-1 infection over more than 10 years of follow-up, and the rest remained negative. More details about this study population are provided in the Methods section and the demographic characteristics are in Table 1.

In this study, we characterized the microbial and metabolic profiles of stool and oral wash specimens as well as the metabolic profile of plasma specimens collected from the initial visit, when all participants were HIV-1 negative. We highlight some of the important findings in the following sections and provide all the findings in Supplementary Data. Greater or lower values in Pre-HIV reported in this study are relative to the Non-HIV group. Unless stated otherwise, results are reported based on $p < 0.01$ and $q < 0.1$, considering $q < 0.05$ as significant and $0.05 \leq q < 0.1$ as suggestive. For differential abundance analysis (DAA), the numbers appearing in parenthesis are natural log fold changes in Pre-HIV relative to Non-HIV.

### Differences in gut microbial composition, functions, and metabolites

We found significantly lower abundances of several gut commensal bacteria, such as *Bacteroides spp.* and *Alistipes spp.*, and significantly higher abundances of proinflammatory bacteria, such as *Holdemanella spp.* and *Prevotella spp.*, in Pre-HIV compared to Non-HIV using shotgun metagenomic data (Fig. 1A and Supplementary Data 1A). These differences in the gut microbial composition in Pre-HIV were accompanied by significant differences in abundances of microbial gene functions (Fig. 1B and Supplementary Data 1B). For instance, enzymes involved in purine catabolism, including allantoinase (0.61) and ureidoglycine aminohydrolase (0.62)[15], and those involved in the citric acid cycle, such as tricarboxylic acid cycle enzyme complex and fumarate metabolic process, were relatively higher in Pre-HIV (0.26). This suggests a shift towards aerobic respiration, potentially due to changes in the oxygen availability in the gut and expansion of facultative anaerobic bacteria, a condition that is associated with inflammation[16,17].

In contrast, several metabolic processes were significantly lower in Pre-HIV, such as amino acid metabolism (e.g., glycine (−0.62), histidine (−0.53), glutamate (−0.72), and tryptophan (−0.13)), urea transport activity (−0.91), which might be an indication of lower utilization of urea as nitrogen source for amino acid biosynthesis, and biosynthesis of trehalose (−0.90) and lipoate (−0.75). There were also functional differences reflective of greater oxidative stress and cell damage in Pre-HIV. For example, response to reactive oxygen species was lower, with lower oxidoreductase acting on peroxide (−0.57) and superoxide dismutase activity (−0.43), which are critical defense mechanisms against oxidative stress, and a greater expression of gene functions related to heat shock protein HslUV (0.49). We detected greater concentrations of hypoxanthine (a purine derivative) (0.88) and thymine (a pyrimidine base) (0.52, $q = 0.36$) in the stool of the Pre-HIV group, further suggestive of a potentially cell damage and DNA degradation (Fig. 1C and Supplementary Data 1C).

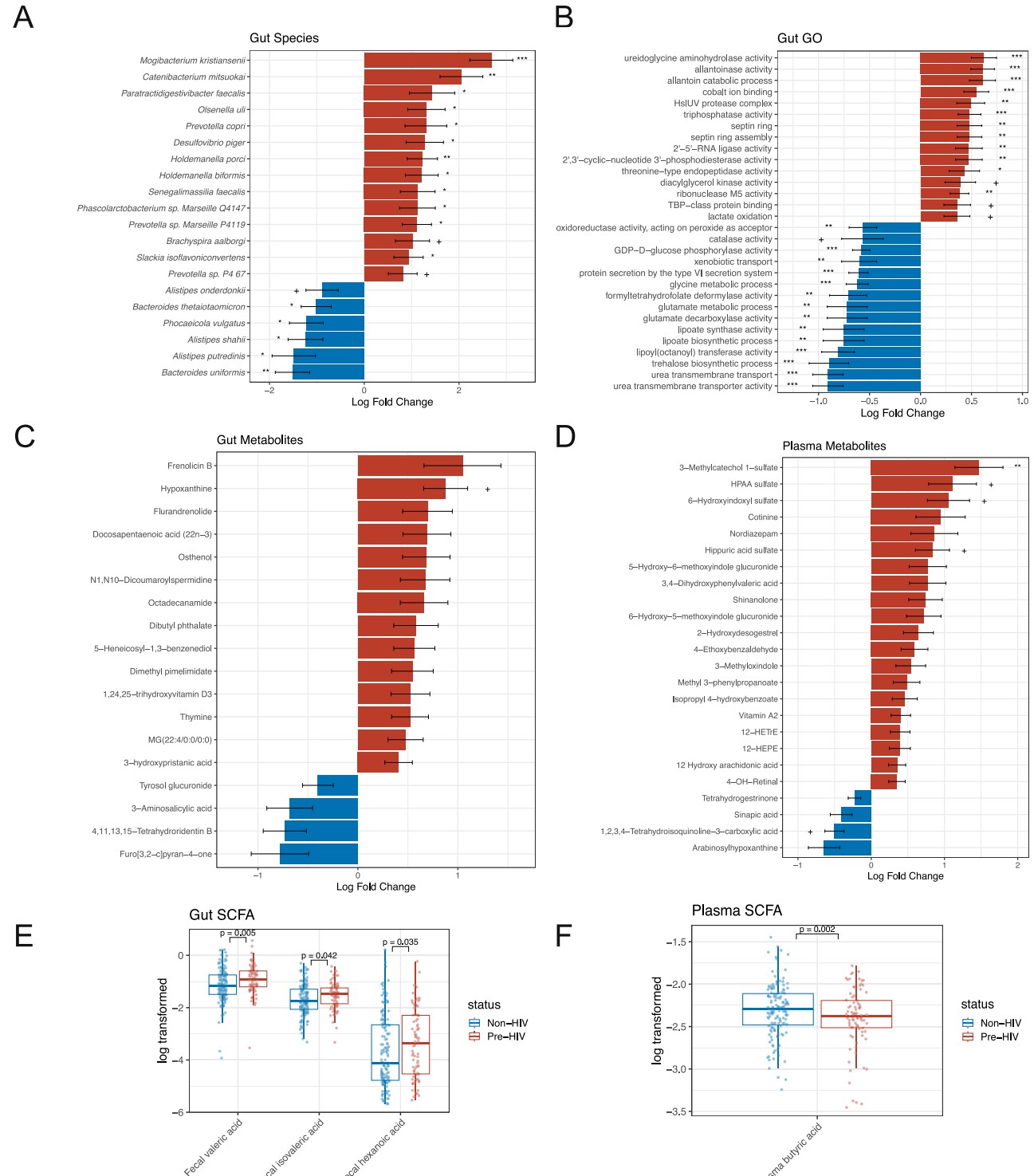

**Fig. 1 | Differential abundance analysis (DAA) of gut microbiome and gut and plasma metabolites between Pre-HIV and Non-HIV. A–D** Log-fold changes of absolute abundances ± SE are presented for differentially abundant features ($p < 0.01$). The red bars to the right of 0 axis correspond to higher abundance in Pre-HIV and blue bars to the left of 0 axis correspond to lower abundance in Pre-HIV. Features that are also significantly different after multiple testing correction using BH procedure are represented by + $q < 0.1$, * $q < 0.05$, ** $q < 0.01$, *** $q < 0.001$. These results are obtained by applying ANCOM-BC2. **A** DAA of bacterial species. **B** DAA of gut microbial Gene Ontology (GO) terms, only the top 30 significant GO terms are shown in this figure. **C** DAA of gut metabolites. **D** DAA of plasma metabolites. **E**, **F** Boxplots of gut and plasma short-chain fatty acids (SCFA). The center line of the boxplot represents the median, the box shows the interquartile range (IQR: Q1–Q3), and the whiskers extend to the most extreme values within 1.5 × IQR from the box. Data points beyond the whiskers are plotted as outliers. The significance of gut and plasma results is based on linear regression analysis. The sample sizes vary by data modality as described in Supplementary Data 12.

Several plasma metabolites with immunomodulatory properties were higher in Pre-HIV (Fig. 1D and Supplementary Data 1D). These included vitamin A2 (0.40, $q = 0.15$) and its derivative 4-OH retinal (0.35, $q = 0.15$). Vitamin A is known to increase the differentiation of CD4$^+$ T cells[18], which could increase CCR5 expression and susceptibility to HIV-1 infection of mucosal CD4$^+$ T cells[19,20]. We also found higher concentrations of eicosanoid derivatives, including 12-HETrE (0.40, $q = 0.22$), 12-HEPE (0.39, $q = 0.36$), and 12-hydroxy arachidonic acid (0.35, $q = 0.20$), also known as 12-hydroxyeicosatetraenoic acid (12-HETE), indicating greater inflammation in Pre-HIV. Several phenyl and indole derivatives exhibited higher plasma concentrations in Pre-HIV, such as hippuric acid sulfate (0.84), 3-methylcatechol 1-sulfate (1.47), 3-methyloxindole (0.54, $q = 0.36$), 6-hydroxyindoxyl sulfate (1.06), and 6-hydroxy-5-methoxyindole glucuronide (0.72, $q = 0.15$), suggesting that either the exposure to aromatic compounds or their metabolism differed before the onset of HIV-1 infection. Indoxyl sulfate metabolites, which are produced from tryptophan by intestinal bacteria and then further metabolized in the liver, are known to be associated with renal dysfunction and vascular diseases[21].

We also observed significantly higher concentrations of gut valeric acid (0.24), isovaleric acid (0.16), and hexanoic acids (0.40) in Pre-HIV (Fig. 1E). Valeric acid is reported to have proinflammatory properties and increase the production of proinflammatory cytokines[22]. As expected, Pre-HIV had significantly lower plasma concentrations of butyric acid (−0.14) (Fig. 1F), suggesting a "leaky gut" phenotype that promotes the translocation of proinflammatory bacteria[23]. Since there were only six SCFAs, these results are based on raw $p < 0.05$.

## Effect of sexual activity on gut microbial composition, functions, and metabolites

The Pre-HIV group had a significantly higher number of receptive anal intercourse partners than the Non-HIV group (Table 1). Since the gut microbiome is significantly associated with sexual orientation[24], it is possible that the observed differences in the gut microbial composition, functions, and metabolites between Pre-HIV and Non-HIV are, in part, due to differences in sexual activity. To understand the extent to which sexual activity affects microbiome and metabolome profiles, we performed a trend analysis of microbial and metabolic features over the four ordered sexual activity groups defined by the number of partners with whom a participant had receptive anal intercourse (Table 1). Overall, we observed that several commensal species belonging to the genera *Alistipes*, *Clostridium*, *Ruminococcus*, and *Streptococcus* decreased with number of receptive anal intercourse partners. In contrast, *Prevotella stercorea*, *Desulfovibrio piger*, and *Mogibacterium kristiansenii* increased with number of receptive anal intercourse partners (Supplementary Fig. 1A and Supplementary Data 2A). Additionally, 54 gene functions, such as those involved in defense mechanisms and oxidoreductase activities, as well as 14 gut metabolites and 17 plasma metabolites, showed a monotonic trend over the four sexual activity groups (Supplementary Fig. 1B–D and Supplementary Data 2B–D).

Comparing the results from DAA between Pre-HIV and Non-HIV with those based on sexual activity groups, we observed that Pre-HIV is associated with more significant changes in gut microbiome composition and function compared to sexual activity. Some of the associations common to both Pre-HIV and sexual activity include an increase in abundances of *Prevotella spp.*, *Mogibacterium kristiansenii*, *Desulfovibrio piger*, and a decrease in *Alistipes spp.* (Supplementary Fig. 2A). Similarly, both Pre-HIV and number of sexual partners seemed to be associated with an increase in allantoinase activity and HslUV protease, and a decrease in lipoate synthase activity, trehalose biosynthetic process, and defense response to gram-negative bacteria (Supplementary Fig. 2B).

Some associations with Pre-HIV appeared unrelated to the number of sexual partners. For example, *Holdemanella biformis*, *Holdemanella porci*, and *Slackia isoflavoniconvertens*, which were higher in Pre-HIV compared to Non-HIV, were not associated with the number of sexual partners (Supplementary Fig. 2A). Additionally, histidine metabolism and protein secretion by the type VI secretion systems were significantly lower in Pre-HIV compared to Non-HIV but were not associated with sexual activity (Supplementary Fig. 2B).

We found a greater number of significant associations between sexual activity and gut and plasma metabolic profiles relative to Pre-HIV, some of which were also associated with Pre-HIV, such as increased plasma phenyl sulfate metabolites in Pre-HIV compared to Non-HIV (Supplementary Fig. 2C, D). Collectively, these results suggest that although sexual activity could be a major driver of changes in the microbial and metabolic composition, some changes in the gut microbiome composition and function preceding HIV-1 infection are potentially unrelated to sexual activity.

## Multi-omics integration of gut microbiome and metabolomes

A multi-omics analysis of the gut microbiome, the corresponding functional terms, and the gut and plasma metabolomes using DIABLO[25] (Fig. 2 and Supplementary Data 3) revealed several notable relationships. For example, among findings with absolute correlation coefficients more than 0.8, microbial conjugated bile acids (microbial bile acid amidates), namely, chenodeoxycholylvaline and chenodeoxycholylleucine, were higher in abundance among Pre-HIV and were negatively correlated with *Oscillibacter sp. MCC667* and *Vescimonas fastidiosa*, whereas were positively associated with proinflammatory eicosanoid derivatives and Vitamin A2 and its derivatives. These associations suggest that alterations in the gut microbiome prior to HIV-1 infection could result in increases in conjugation of bile acids with amino acids, affecting both the bile acids pool and amino acid concentrations in the gut. Indeed, recent studies showed that these microbial bile acid amidates interact with host via nuclear receptors like the farnesoid X receptor and pregnane X receptor[26,27], and may be implicated in inflammatory bowel disease[27] by mediating interferon-γ production in CD4$^+$ T cells[27].

Additionally, we found that the increased abundance of *Mogibacterium kristiansenii* in Pre-HIV was positively associated with plasma phenyl and indoxyl/indole sulphates as well as 2,8-Dihydroxyquinoline-betaglucuronide, which is the glucuronide conjugate of 2,8-Dihydroxyquinoline, an intermediate in gut microbial metabolism of quinoline derivatives[28,29].

## Differences in gut microbial interactions

In addition to multi-omics analysis, we examined differences between Pre-HIV and Non-HIV in correlations among gut microbial species (Supplementary Data 4A), and correlations between gut microbial species and various data modalities, namely, gut microbial functional terms (Supplementary Data 4B), gut metabolome (Supplementary Data 4C, D), and plasma metabolome (Supplementary Data 4E, F). Pairs of features with the absolute difference in correlation more than 0.3, $p < 0.01$, and $q < 0.1$ are reported, considering $q < 0.05$ as significant and $0.05 \leq q < 0.1$ as suggestive (Supplementary Data 5). To further assess whether the differentially correlated features were related to sexual activity, we performed a trend analysis of the pairs of features over the four sexual activity groups (Supplementary Data 5). Depending on the data modality, about 8% to 20% of differential correlated features between Pre-HIV and Non-HIV with $q < 0.1$ exhibited a decreasing or increasing trend over the four sexual activity groups at $p < 0.01$ and $q < 0.1$, suggesting that sexual activity contributed to some of the differences in correlations among features between Pre-HIV and Non-HIV. Here, we present only a subset of significant differential correlations that could potentially be biologically relevant (Fig. 3).

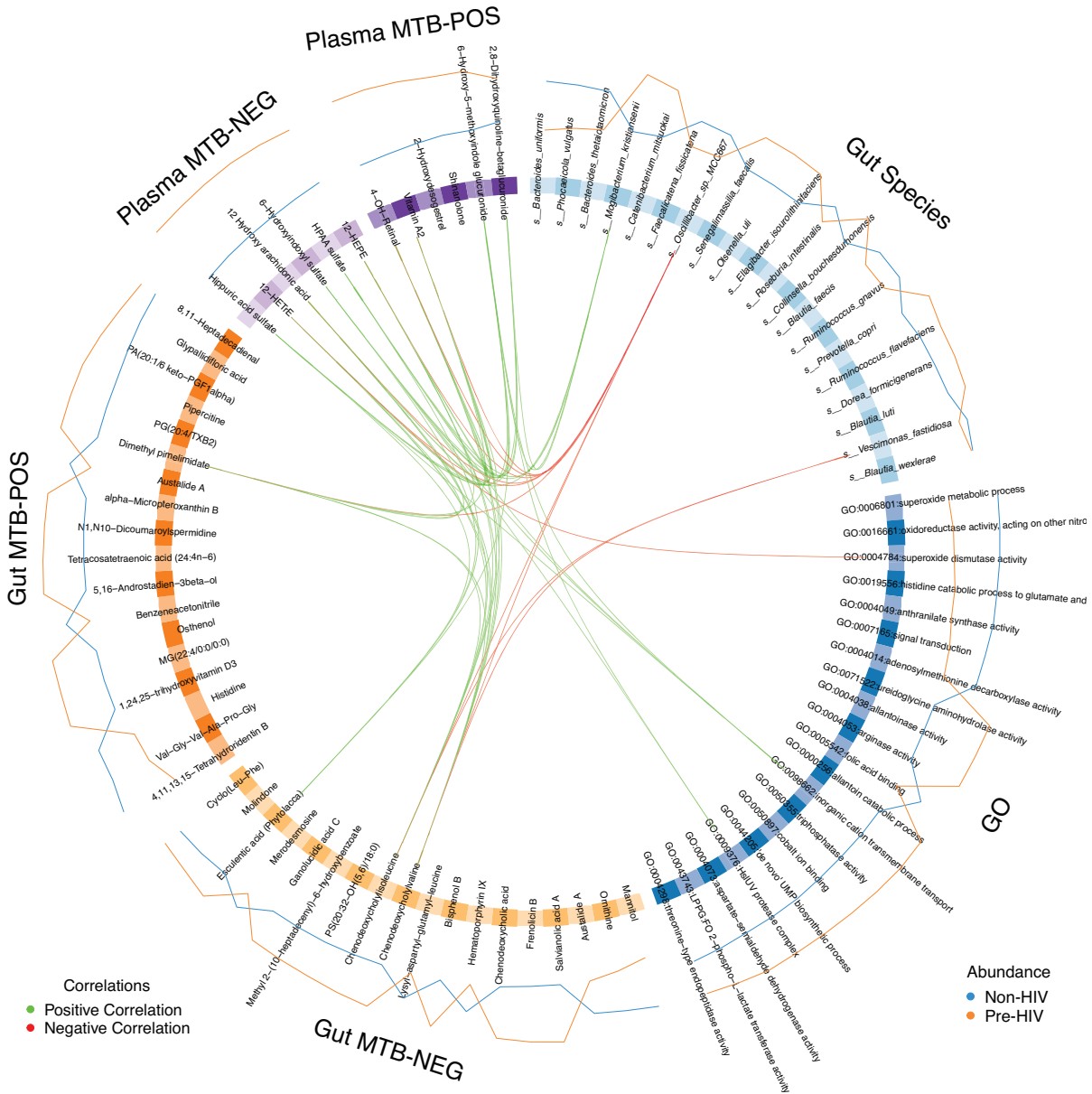

**Fig. 2 | Multi-omics analysis.** Circos plot showing integrative multi-omics features from gut species, gut microbial Gene Ontology (GO) terms, gut metabolites from positive ion channel (Gut MTB – POS), gut metabolites from negative ion channel (Gut MTB – NEG), plasma metabolites from positive ion channel (Plasma MTB – POS), and plasma metabolites from negative ion channel (Plasma MTB – NEG) that are associated with Pre-HIV. These results are obtained by applying DIABLO. Features from each data modality are shown on the circle. Green and red lines connecting the features indicate positive and negative correlations, respectively. Only correlations with absolute correlation coefficient more than 0.8 are shown in the figure. The lines outside of the circle represent abundances of features in Pre-HIV and Non-HIV. Only features identified from latent components 1 and 2 are shown. For this analysis, samples that have all the three data modalities (gut microbiome, gut metabolites, and plasma metabolites) were included (Pre-HIV $n = 82$, Non-HIV $n = 148$).

We observed that the correlations between proinflammatory species *Holdemanella biformis*, *Holdemanella porci*, and several commensal bacteria, such as *Pseudoflavonifractor sp. MSJ_30*, *Oscillibacter sp. ER4*, *Oscillibacter sp. MSI 31*, and *Intestimonas sp. MSJ 38* were more negatively correlated in Pre-HIV compared to Non-HIV (Fig. 3A and Supplementary Data 5A). Interestingly, these relationships seemed to be not related to sexual activity since the correlations between *Holdemanella spp.* and the commensal bacteria did not have any increasing or decreasing pattern over the four sexual activity groups (Supplementary Fig. 3A). These results are consistent with our DAA, where *Holdemanella spp.* were significantly higher in Pre-HIV but were

not associated with sexual activity. Overall, these results suggest that higher abundances of potentially proinflammatory *Holdemanella spp.* are correlated with lower abundances of commensal bacteria prior to HIV-1 infection, which could be the consequence of the gut dysbiosis.

Relationships between gut species and functional terms related to de novo purine and pyrimidine biosynthesis, including carbamoyl-phosphate synthase (glutamine-hydrolyzing) activity (GO:0004088), an enzyme involved in the first step of pyrimidine biosynthesis, and phosphoribosylformylglycinamidine cyclo–ligase activity (GO:000 4641), an enzyme that catalyzes the formation of the imidazole ring in the de novo purine biosynthesis pathway, were different in Pre-HIV

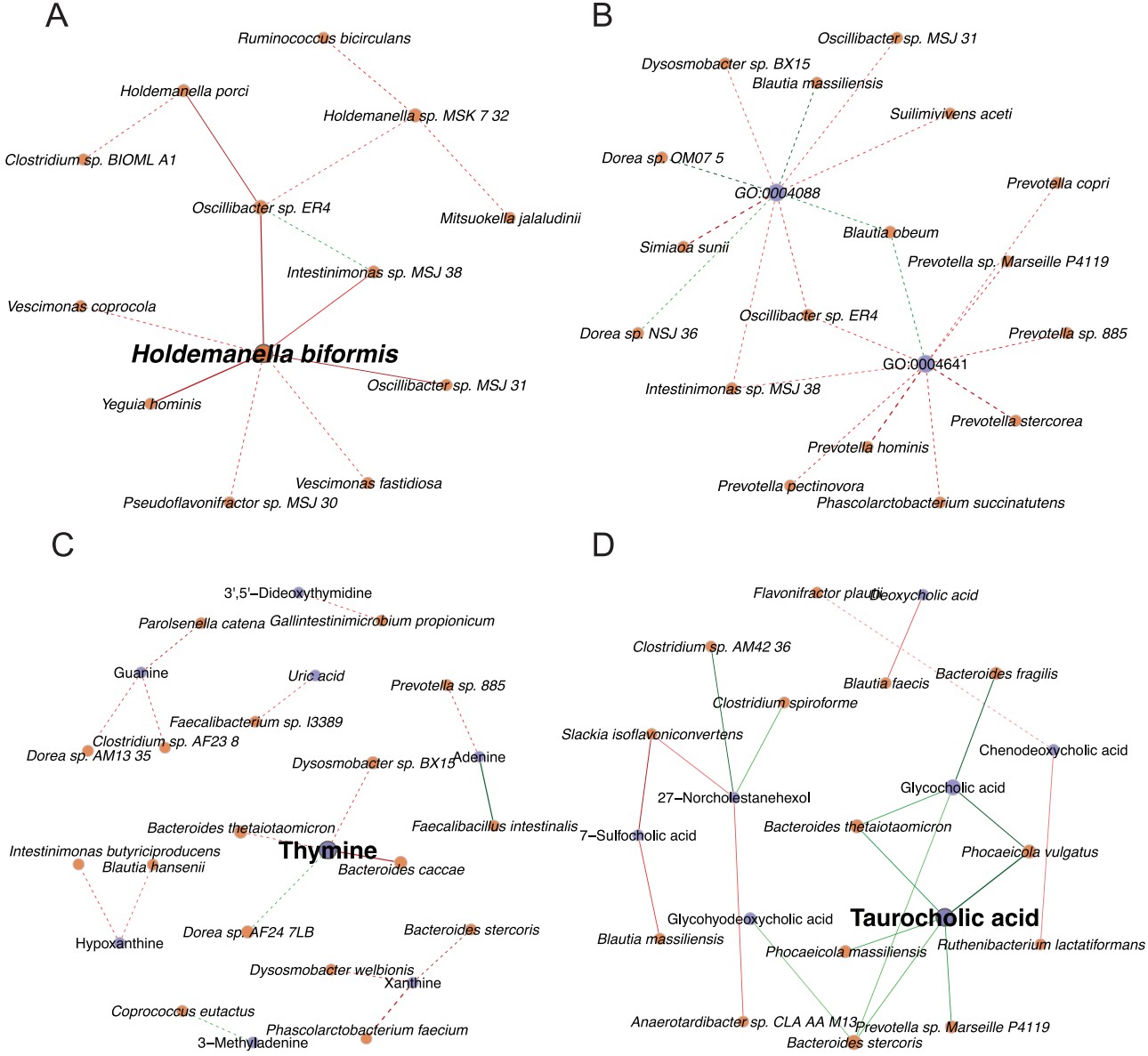

**Fig. 3 | Differential correlation analysis related to gut microbiome between Pre-HIV and Non-HIV.** Networks show a subset of significant differential Spearman correlations between Pre-HIV and Non-HIV with absolute difference in correlation > 0.3, $p < 0.01$, and $q < 0.1$. **A** Differential Correlations between *Holdemanella spp.* and other gut species. **B** Differential correlations between Gene Ontology (GO) terms related to enzymes involved in purine and pyrimidine biosynthesis pathways (GO:0004641 and GO:0004088, respectively) and gut species. **C** Differential correlations between gut purine and pyrimidine derivatives and gut species. **D** Differential correlations between plasma bile acids and gut species. Within each network, a pair of dots (nodes) is connected by an edge if the pair is differentially correlated between Pre-HIV and Non-HIV with $p < 0.01$, $q < 0.1$, and the absolute difference in correlation more than 0.3. Correlations with $q < 0.05$ and $0.05 \le q < 0.1$ are presented by solid lines and dashed lines, respectively. Green edges correspond to correlations that are higher in Pre-HIV compared to Non-HIV, and red edges correspond to correlations that are lower in Pre-HIV compared to Non-HIV. Brown nodes represent gut species, and purple nodes represent GO terms or metabolites. All Spearman correlation coefficients are derived after correcting for compositionality and adjusting for age, location, and antibiotic use as covariates. Correlation coefficients were corrected for different sample sizes of Pre-HIV and Non-HIV and were compared using two-sided Fisher's $Z$ test. The sample sizes vary by data modality as described in Supplementary Data 12.

versus Non-HIV (Fig. 3B and Supplementary Data 5B). Especially, these two enzyme activities were more negatively correlated with *Oscillibacter sp. ER4*, while more positively correlated with *Blautia obeum* in Pre-HIV compared to Non-HIV. Furthermore, several *Prevotella spp.* and *Phascolarctobacterium succinatutens* were all negatively correlated with phosphoribosylformylglycinamidine cyclo–ligase activity in Pre-HIV. Most of these relationships had a monotonic pattern over the four sexual activity groups (Supplementary Fig. 3B). Collectively, changes in the correlations between these species and purine and pyrimidine biosynthesis in the gut precedes HIV-1 infection, which could be related to sexual activity.

Several pyrimidine and purine derivatives were differentially correlated with gut species between Pre-HIV and Non-HIV. In particular, thymine, a pyrimidine base tended to be higher in Pre-HIV, was negatively correlated with *Bacteroides caccae, Bacteroides thetaiotaomicron*, and *Dysosmobacter sp. BX15*, whereas positively correlated with *Dorea sp. AF24 7LB* and *Holdemanella biformis* ($q = 0.11$, Supplementary Data 4D) in Pre-HIV. In addition, xanthine and hypoxanthine (purine derivatives) were differentially negatively correlated with several commensal gut bacteria, such as *Bacteroides stercoris* and *Intestinimonas butyriciproducens*, respectively (Fig. 3C and Supplementary Data 5D), while were positively correlated with *Holdemanella*

*biformis* in Pre-HIV ($q = 0.11$ for xanthine and $q = 0.20$ for hypoxanthine, Supplementary Data 4D). A decreasing correlation between thymine and *Bacteroides caccae* and *Dysosmobacter sp. BX15* and an increasing correlation between thymine and *Dorea sp. AF24-7LB* were also observed over the four sexual activity groups (Supplementary Fig. 3C).

Interestingly, we found that histamine, an immune stimulator, was negatively correlated with several commensal bacteria, such as *Ruminococcus champanellensis, Ruminococcus flavefaciens*, and *Clostridium sp. BIOML A1* in Pre-HIV (Supplementary Data 5D), while it was positively correlated with *Holdemanella biformis* ($q = 0.11$) and *Holdemanella porci* ($q = 0.12$) in Pre-HIV (Supplementary Data 4D). Furthermore, relationships between gut species, such as *Holdemanella biformis, Yeguia hominis, Pseudoflavonifractor sp. MSJ 30*, and sphingosine(1+), a lipid immune modulator, were also differentially correlated between Pre-HIV and Non-HIV (Supplementary Data 5D). Except for *Holdemanella biformis*, the correlations between sphingosine(1+) and gut species seem to be mostly associated with sexual activity (Supplementary Data 5D).

Differential networks based on plasma metabolites and gut species indicate that interactions between plasma bile acids and gut species were significantly different between the Pre-HIV and Non-HIV groups. We found positive correlations between glycocholic acid and taurocholic acid with *Bacteroides stercoris, Bacteroides fragilis, Bacteroides thetaiotaomicron*, and with *Phocaeicola vulgatus* and *Phocaeicola massiliensis* in Pre-HIV (Fig. 3D, and Supplementary Data 5E). Since *Bacteroides spp.* have reduced abundance in Pre-HIV, the differential correlations suggest that disruption in *Bacteroides* in Pre-HIV is associated with changes in plasma bile acids. The correlations between gut species and plasma bile acids did not change significantly over the sexual activity groups (Supplementary Fig. 3D).

### Oral microbiome dysbiosis

We repeated the above analyses for the oral microbiome and metabolome. Although we observed fewer differences between Pre-HIV and Non-HIV in the oral microbiome and metabolome relative to those from the gut, we observed that several oral proinflammatory bacteria tended to be higher in Pre-HIV versus Non-HIV. We also observed that the interactions among oral species as well as between oral species and oral gene functions, oral metabolome, plasma metabolome, and gut species were significantly different between Pre-HIV and Non-HIV. All the findings and their interpretations are included in the Supplementary Information, Supplementary Figs. 4–8 and Supplementary Data 6–9.

### Measurement of disruption in correlations

Based on differential correlations analysis, we quantified taxon-specific disruptions in correlations using DISCO as described in the "Methods" section. DISCO identified clusters of gut and oral species that are disrupted in their correlations depending on the type of data modality (Fig. 4A, B and Supplementary Data 10). There is an orthogonality in both gut and oral heatmaps where some species were exclusively disrupted in Pre-HIV in their correlations with other species or the GO terms, but not gut/plasma metabolites. Examples of such gut species include *Mogibacterium kristiansenii, Bacteroides xylanisolvens, Bacteroides thetaiotaomicron*, and *Blautia faecis* (Fig. 4A). Examples of oral species that were disrupted with other oral species or their gene functions included some species within the genera *Rothia* and *Streptococcus* as well as *Candidatus Nanosynbacter TM7* (Fig. 4B). On the other hand, some species were exclusively disrupted in Pre-HIV in their correlations with metabolites and not with other species. For example, in the case of gut species, correlations of *Eggerthella lenta* and *Slackia isoflavoniconvertens* were exclusively disrupted with gut and plasma metabolites (Fig. 4A). Interestingly, correlations of oral *Porphyromonas spp.* (*P. bobii, P. somerae*), *Actinomyces spp.*, *Streptococcus spp.* (such as

*S. anginosus*), and *Prevotella spp.* (*P. salivae, P. pallens, P. melaninogenica*) appeared not to be disrupted with other oral species but disrupted with gut species (Fig. 4B). Interestingly, among these oral species, correlations of *Prevotella spp. and Streptococcus spp.* were also disrupted with plasma metabolites (Fig. 4B).

To evaluate the performance of DICSO, we applied it to four external datasets that included participants with or without HIV-1. Details are provided in the Supplementary Information, Supplementary Figs. 9 and 10 and Supplementary Data 11. Notably, these results showed that *Prevotella spp.* were disrupted in their correlations prior to or following HIV-1 infection across most cohorts. Overall, the results from our study as well as other studies highlight that not only changes in abundances of bacteria could be indicative of a dysbiotic state, but also changes in their interactions across different data modalities could drive the dysbiosis preceding HIV-1 infection or following the HIV-1 infection.

## Discussion

The primary goal of this study was to understand system-level changes before HIV-1 infection that may help in developing prevention and/or treatment interventions. Our analyses confirmed recent findings on the gut microbiome and revealed several novel insights as summarized below.

### Commensal bacteria

We found several commensal bacteria, such as *Alistipes* and *Bacteroides*, to be lower in abundance prior to HIV-1 infection. *Alistipes spp.* have a symbiotic role with the host and a protective role against inflammation and various diseases, including HIV-1 infection[30–33]. Most species within the genus *Alistipes* produce anti-inflammatory metabolites such as SCFA from fermenting carbohydrates and indole from metabolizing tryptophan[30]. Several species of *Bacteroides* contribute to production of the anti-inflammatory IL-10 and modulate T regulatory cells (Tregs)[34]; therefore, their reduction prior to HIV-1 infection could contribute to host immune activation and increased susceptibility to HIV-1 infection. We found that many of *Bacteroides spp.* were significantly differentially correlated with gene functional terms and metabolites between Pre-HIV and Non-HIV. For instance, in Pre-HIV, *Bacteroides xylanisolvens* was positively correlated with response to oxidative stress and glutathione peroxidase activity, which plays a crucial role in protecting cells from oxidative damage. Therefore, it is possible that a lower abundance of *Bacteroides xylanisolvens* in Pre-HIV was associated with a lower response to oxidative stress. We also found that *Bacteroides thetaiotaomicron* was positively correlated with plasma glycocholic acid and taurocholic acid in Pre-HIV. This relationship could be explained by the fact that *Bacteroides thetaiotaomicron* is capable of metabolizing bile acids via bile salt hydrolase or hydroxysteroid dehydrogenase in the gut, further inducing synthesis of primary bile acids in the liver via a negative feedback loop[35]. The reason that these relationships were observed in Pre-HIV and not in Non-HIV could be due to the reduction in the abundance and higher variation of this species in Pre-HIV.

### Proinflammatory bacteria

We found several putatively proinflammatory species higher in abundance prior to HIV-1 infection. Some of them, such as *Prevotella*, *Catenibacterium*, and *Holdemanella* species, were previously reported to be associated with sexual activity independent of HIV-1 status and more prevalent in MSM[4]. Since we observed higher abundances of these microbes in MSM prior to HIV-1 acquisition compared to MSM participants that did not become HIV-1 infected, we speculate that higher abundances of these microbes could contribute to increased susceptibility to HIV-1 infection. This is supported by Yamada et al.[36], who found that *Holdemanella biformis* and *Catenibacterium mitsuokai* were positively correlated with the frequency of CCR5+ expressing

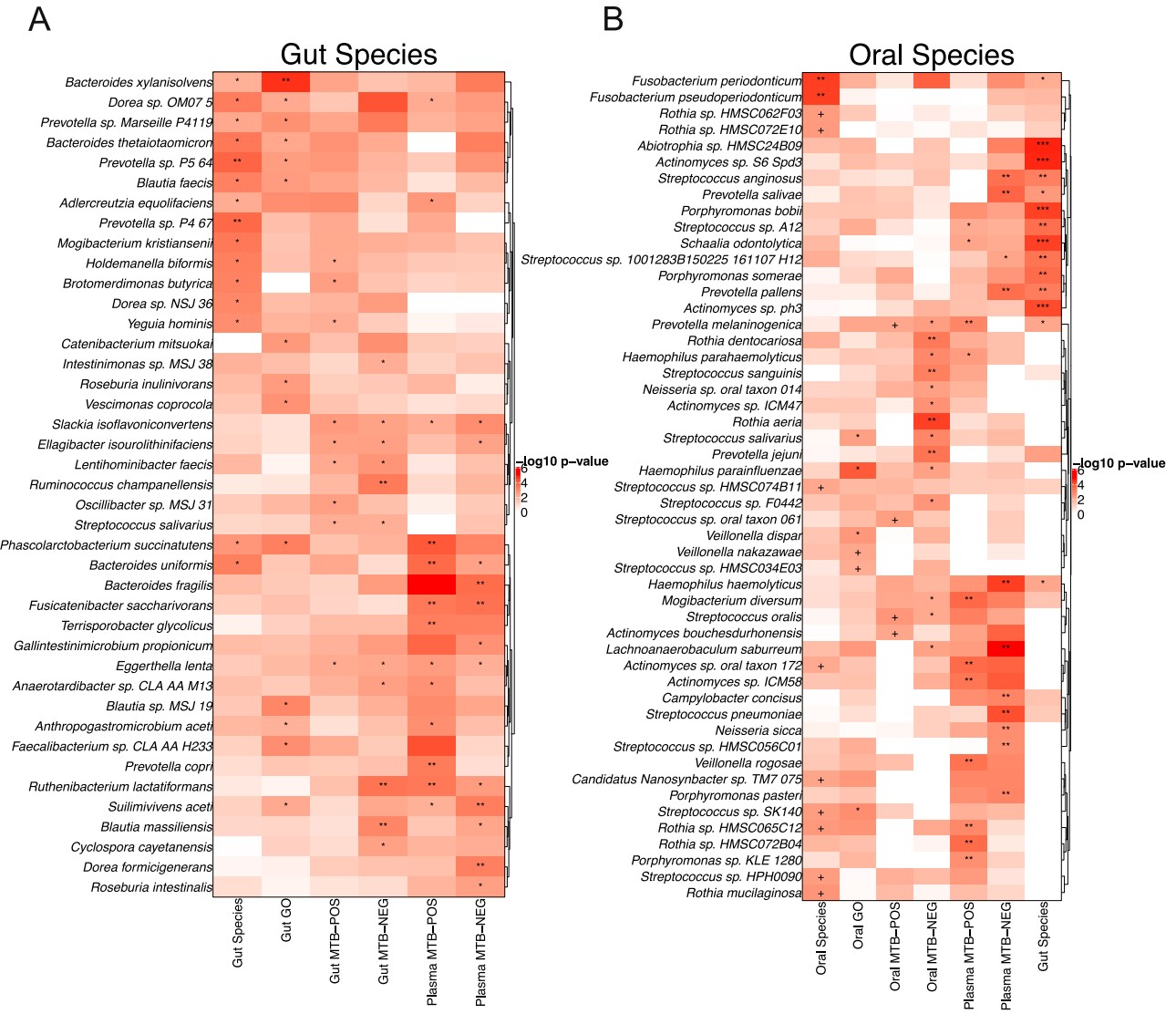

**Fig. 4 | Heatmaps of DISCO.** In each panel, species with significant DISCO score in each data modality are denoted by + $p < 0.01$ and $q < 0.1$, * $p < 0.01$ and $q < 0.05$, ** $p < 0.01$ and $q < 0.01$, *** $p < 0.01$ and $q < 0.001$. The $p$-values are derived from standard Cauchy distribution. The number of correlations for each significant species is more than 20% of maximum correlations in a data modality. For each data modality the top 10 most significant species were selected for graphical representation. **A.** gut species. **B.** oral species. MTB-POS and MTB-NEG represent metabolites from positive and negative ion channels, respectively. The sample sizes vary by data modality as described in Supplementary Data 12.

CD4[+] T cells, in human gut biopsy samples from MSM with and without HIV-1. In their in vitro models, these bacteria modulated the expression of CCR5 by inducing inflammatory cytokines, resulting in a higher ratio of tumor necrosis α (TNF-α) to IL-10 [36]. These microbes were not only observed in higher abundances in PWH but were also reported to be enriched in other inflammatory disease, further suggesting that they could play a role in elevating inflammation and immune responses[37–39]. Interestingly, based on our differential correlation analysis, *Holdemanella biformis* had a high DISCO score, suggesting that disruption in the correlations of this species occurs prior to HIV-1 infection. For example, *Holdemanella spp.* were negatively correlated with several commensal bacteria in Pre-HIV. We also found *Holdemanella spp.* in Pre-HIV to be positively correlated with proinflammatory metabolites in the gut, including histamine and sphingosine(1+), which is a lipid immunomodulator[40]. Additionally, *Holdemanella biformis* was associated with higher levels of thymine in the gut, which could be indicative of higher DNA degradation and cell damage in the gut. Trend analysis of *Holdemanella spp.* over sexually active groups showed that changes in abundances and correlations of *Holdemanella spp.* were not

related to the number of sexual partners. All these results point to the proinflammatory role of *Holdemanella spp.* and their potential contribution to the increased risk of HIV-1 infection.

### Amino acid and purine metabolism

Our results from both metagenomics and metabolomics analyses suggest that amino acid metabolism changes before HIV-1 infection. In particular, the microbial tryptophan biosynthesis pathway was depleted in Pre-HIV, while phenyl and indole metabolites, the microbial metabolites of aromatic amino acid catabolism[41,42], were enriched in Pre-HIV. Additionally, higher concentrations of the fecal branched chain fatty acid (isovaleric acid), which is the major byproduct of amino acid metabolism[43], suggest a shift from amino acid biosynthesis to amino acid catabolism prior to HIV-1 infection. These observations are consistent with previous studies showing HIV-1 infection is associated with changes in plasma metabolites related to amino acid metabolism[44–48]. Nystrom et al. reported perturbed glycine and serine metabolism pathways and lower plasma tryptophan and its metabolites in PWH compared to PWOH[49]. Serrano-Villar et al. showed HIV-1

infection is associated with impaired gut microbiome capacity for production of amino acids, including proline, phenylalanine, and lysine, compared to other inflammatory diseases and healthy controls[50].

We observed that purine metabolism was dysregulated prior to the onset of HIV-1 infection. *Phascolarctobacterium succinatutens* and *Prevotella spp.*, higher in Pre-HIV than Non-HIV, were associated with decreased microbial capacity of purine biosynthesis. In addition, the metabolic profile before HIV-1 infection shifted toward purine degradation as the abundance of hypoxanthine and genes that encode degradation of purine derivatives were enriched in Pre-HIV. The end product of purine metabolism is uric acid, which exacerbates inflammation and is implicated in inflammatory diseases[51–53]. Given the inflammatory end products of purine, an increase in purine metabolism prior to HIV-1 infection could contribute to inflammation.

## Oxidative stress

The metabolic profile of gut microbiome prior to HIV-1 infection revealed reduced capacity in the production of biomolecules that are essential to protect against oxidative stress. For example, biosynthesis of trehalose and lipoate were reduced in Pre-HIV. Trehalose is a disaccharide of D-glucose with structural and functional roles in bacteria; it protects bacteria against abiotic stress and potentially oxidative stress[54] by protecting proteins and unsaturated fatty acids from oxidative damage[55]. Lipoate is an organosulfur cofactor essential to the function of several enzyme complexes involved in oxidative and one-carbon metabolism. Due to its redox activity, it is considered as an antioxidant and free-radical scavenger, thereby protecting against harmful free radicals[56,57]. Lipoic acid has been shown to be an effective HIV-1 inhibitor[58], and thus, the reduction in its biosynthesis by the gut microbiome could potentially increase HIV-1 replication. Besides the alterations in production of these protective biomolecules, the capacity of the microbiome to produce heat shock protein, which occurs in response to stress, was higher in Pre-HIV. Several plasma proinflammatory metabolites, such as eicosanoid derivatives, which are known to be elevated in PWH[59], were also higher in Pre-HIV. Interestingly, our multi-omics integrated analysis showed that eicosanoid derivatives were associated with gut microbial conjugated bile acid, a recently discovered class of bile acids that were associated with IBD and cytokine production[27]. On the other hand, the anti-inflammatory plasma butyric acid was lower in the Pre-HIV. Overall, these results suggest that changes in the gut microbiome and increased mucosal inflammation prior to infection are associated with higher oxidative stress. These results are in line with findings from other studies that have shown HIV-1 infection is associated with changes in redox potential and increased oxidative stress[44,60,61].

## DISCO

The novel methodological contribution of this study is the introduction of a new measure of disruption in correlations for each individual species that characterizes the importance of inter-species interactions, as well as microbial species interactions with microbial functions and metabolites. DISCO is based on the deviation of interaction of microbiome with other biological components in a disease group relative to a control group. In our study, DISCO showed that several gut and oral microbial species, such as gut species belonging to *Holdemanella, Bacteroides, Prevotella,* and *Blautia,* as well as oral species belonging to *Fusobacterium, Rothia, Prevotella,* and *Porphyromonas,* were system-wide disrupted in their interactions with other microbial species, their gene functions, or/and metabolites in Pre-HIV compared to Non-HIV. Application of DISCO to external datasets revealed that although several genera were uniquely disrupted to each cohort, reflecting the inherent differences between cohorts, such as different demographics, geographical location, sexual orientation, etc., *Prevotella spp.* were consistently disrupted in their correlations either before or following

HIV-1 infection across multiple cohorts. These results highlight the role of *Prevotella spp.* in the pathogenesis of HIV-1 infection.

## Strengths and limitations

This study is based on a unique 40-year-old cohort when none of the participants were HIV-1 positive at the time of specimen collection and did not receive any treatments for HIV-1 infection. Hence, a strength of this study is that we can rule out the effects of HIV-1 treatments on microbial dysbiosis findings of this paper. However, due to the unavailability of data regarding diet, we were unable to investigate the effects of diet on our findings. In particular, the effect of diet and other unmeasured confounders on DISCO is unknown and requires further investigation in future studies. Additionally, the present study is an association study and does not provide evidence for causality. Although we have provided mechanistic insights with the analysis of GO terms and metabolomes, in vitro and/or in vivo studies will be required to validate our findings.

In summary, our study showed a system-level difference between MSM who acquired HIV-1 infection and those who did not. Greater oxidative stress and differences in amino acid and purine metabolism preceded HIV-1 infection. Interactions within species and between species and metabolites were significantly different between these two groups of MSM, suggesting that microbial ecological differences exist before HIV-1 infection and potentially contribute to the increased risk of HIV-1 infection. Our study revealed links between gut and oral microbes with metabolites, which could be subject to further mechanistic studies to untangle these relationships and their roles in acquisition of HIV-1 infection.

# Methods

## Ethics and Institutional Review Board (IRB)

The MACS samples and data used in this study are based on research that was conducted with institutional review board approvals from all participating institutions: Johns Hopkins School of Public Health, University of Pittsburgh School of Public Health, Northwestern University School of Medicine, and UCLA School of Public Health. All ethical regulations relevant to human research participants were followed and approved by all participating institutions' Review Boards, and informed consent was obtained from all study participants.

## Study design and data acquisition

This study included 239 MSM enrolled in the MACS between 1984 and 1985 in four US cities: Pittsburgh ($n = 43$), Baltimore-Washington DC ($n = 63$), Chicago ($n = 64$), and Los Angeles ($n = 69$). All 239 participants were HIV-1-negative at their first MACS clinic visit as defined by lack of HIV-1 ELISA and Western blot antibodies, and HIV-1 RNA by rtPCR. Of these 239 participants, 87 were defined as being primary HIV-1 infected by testing positive for serum HIV-1 enzyme immunoassay and Western blot antibodies and confirmed positive for HIV-1 RNA by rtPCR, at their second MACS clinic visit approximately 6 months later. These 87 men were termed Pre-HIV-Infection (Pre-HIV), while the other 152 participants remained HIV-negative at their second MACS clinic visit (Non-HIV). The stool, oral wash, and plasma samples obtained from participants during their first MACS clinic visit were analyzed in this study for microbiome, metabolome, and SCFA. Information on pertinent covariates was obtained from original study questionnaires and included age, MACS site, and antibiotic use, which were adjusted in all the analyses performed in this study.

The DNA was extracted from the stool and oral wash samples using the PowerSoil DNA Extraction Kit (MO BIO Laboratories, Carlsbad, CA). The libraries for metagenomics sequencing were constructed from the extracted DNA and sequenced by the NCI LICI Microbiome and Genetics Core (NCI Bethesda). The stool and oral wash microbiome data were obtained using shotgun metagenomics with a target sequencing depth of 20 million reads. Illumina paired

fastqs for each sample were processed using the JAMS package version 1.9.7[62], available at https://github.com/johnmcculloch/JAMS_BW. Sequencing reads for each sample were processed, in parallel, using the JAMSalpha script, which involved quality trimming and adapter clipping of raw reads with Trimmomatic 0.36[63], followed by alignment against the human genome with Bowtie2 v2.3.2[64]. Following quality trimming and removal of host reads, the average number of reads was 6.07 million and 24.0 million for oral and gut samples, respectively. Non-host reads were assembled of using MEGAHIT v1.2.9[65]. Assembly contigs larger than 500 bp were taxonomically classified with Kraken2[66], using a custom 196-Gb Kraken2 database including draft and complete genomes of all bacteria, archaea, fungi, viruses, and protozoa available in the NCBI GenBank in December 2022, in addition to human and mouse genomes, using the JAMSbuildk2db tool of the JAMS package. Functional annotation of contigs was obtained using Prokka v1.14.6[67]. The predicted proteome from the contigs of each sample were further functionally classified using InterProScan (https://github.com/ebi-pf-team/interproscan). GO annotations from Inter-ProScan were used for down-stream statistical analyses. The sequencing depth of each contig was obtained by aligning the reads used for assembly back to the contigs. Taxonomic features were expressed as the last known taxon (LKT), which is the lowest taxonomically unambiguous classification determined for each query sequence. Feature-by-samples counts tables were generated and stored into SummarizedExperiment objects (https://bioconductor.org/packages/SummarizedExperiment), which in turn were constructed using the JAMSbeta pipeline of the JAMS package.

Metabolomics and SCFA were measured by Creative Proteomics, Inc. (NY, USA, www.creative-proteomics.com), and details are provided in the Supplementary Methods. Annotation of metabolites were obtained using The Human Metabolome Database[68]. The number of samples across data modalities varied slightly as we were not able to generate data from a few low-quality samples. Number of samples for each data modality is included in Supplementary Data 12.

## Statistical analyses

**Data filtering.** Lowest known Taxa (LKT) present in more than 10% of samples with a relative abundance of at least 50 ppm (parts per million) and genome completeness of at least 10% were retained. For untargeted metabolomics, annotated metabolites with an error of 5 ppm or more were removed. 10% of metabolites with the lowest relative standard deviation were also removed.

**Differential abundance analysis (DAA) and trend analysis.** All microbiome analyses were performed at the species level. DAA and trend analysis were performed using ANCOM-BC2 v.2.6.0[69]. DAA was performed to identify differentially abundant species between Pre-HIV and Non-HIV, and trend analysis was performed to identify species that have a decreasing or increasing trend over sexual activity groups defined by the number of partners with whom a participant had receptive anal intercourse ($G_1$: 0, $G_2$: 1 $G_3$: 2–5, and $G_4$: 6 or more partners).

For microbiome data the prevalence cut-off was set to 20% and a pseudo-count sensitivity test, as described in ref. 69, was performed for each species. ANCOM-BC2 was also used for performing DAA of gene ontologies (GO) as well as metabolomics data, since these data are considered as compositional. Similar to taxonomic data, GO terms are compositional because they are obtained from high-throughput sequencing[70]. Mass spectrometry-based metabolomics data acquired by ion trap mass spectrometers are also inherently compositional. The number of ions present in any given spectrum is a proportion of a targeted maximum (e.g., 1 million ions), which is a compromise between sensitivity and minimization of deleterious space charging effects. For this reason, compositionally aware methods are recommended for normalizing untargeted metabolomics in the literature[71,72].

For metabolomics, we did not have to invoke pseudo-count sensitivity test when analyzing metabolomics data due to data imputation (Supplementary Methods). For all differential abundance analyses, age, location, and antibiotic use were included in the model as covariates. Significance of results were determined by a $p$-value threshold of 0.01. Features that survived multiple testing correction using Benjamini–Hochberg Procedure (BH) procedure at $q < 0.05$ were considered as significant, while those with $0.05 \leq q < 0.1$ were considered as suggestive. These results are represented by + ($q < 0.1$), * ($q < 0.05$), ** ($q < 0.01$), ***($q < 0.001$) in the figures.

For SCFAs, univariate linear regression models with age, location, and antibiotic use as covariates were used to compare SCFA concentrations between Pre-HIV and Non-HIV groups. SCFA analyses were performed using complete data.

**Multi-omic integration.** Data Integration Analysis for Biomarker discovery using Latent cOmponents (DIABLO)[25] from the R package of mixOmics v.6.28.0 was used for integration of gut microbiome, gut metabolites, and plasma metabolites. For this analysis, samples that have all the three data modalities (gut microbiome, gut metabolites, and plasma metabolites, Pre-HIV $n = 82$, Non-HIV $n = 148$), were included. Normalized and bias-corrected data obtained from ANCOM-BC2 were used for multi-omics integration analysis. Furthermore, to remove the confounding effects of age, location, and antibiotic use, residuals obtained from linear regression models, including those covariates, were used as inputs for DIABLO. A similar workflow as described in https://mixomics.org/mixdiablo/diablo-tcga-case-study/ was used for our analysis. A value of 0.1 was used in the design matrix. A model was trained using 10 cross-validation with 10 repeats.

**Differential correlation analysis.** All pairwise correlations between features were performed using Spearman rank-order correlation after normalizing the data using central log-ratios to address compositionality based on SECOM methodology[73]. To control confounding factors, normalized data were fitted to linear models with age, location, and antibiotic use as independent variables and the residuals for these models were used in the correlation analyses. To reduce spurious results due to sparsity, only features present in more than 50% of samples were included in correlation analyses. Within each group (Pre-HIV and Non-HIV), Spearman correlations among species as well as between species and gene functions (GO terms) and metabolites were computed using complete data. Let $\rho_{ijk}$ denote the population Spearman correlation coefficient between a pair of features $X_{ik}$ and $X_{jk}$ in the $k^{th}$ group, $k = 1, 2$, and let $\hat{\rho}_{ijk}$ denote the corresponding sample estimator. We denote the corresponding Fisher's Z-transformed correlation coefficients and the corresponding bias corrected estimators by $\theta_{ijk} = \tanh^{-1}\rho_{ijk}$ and $\hat{\theta}_{ijk} = \tanh^{-1}\hat{\rho}_{ijk} - \frac{\hat{\rho}_{ijk}}{2(n_{ijk}-1)}$, respectively, where $n_{ijk}$ is the number of samples corresponding to feature pair $(i,j)$ in the $k^{th}$ group, $k = 1, 2$. To deal with sparsity, we performed hard thresholding to filter out small correlation coefficients. Specifically, for each $i, j$, for $k = 1, 2$, we tested $H_0 : \theta_{ijk} = 0$ vs. $H_a : \theta_{ijk} \neq 0$ by invoking $\hat{\theta}_{ijk} \sim^{asymp} N\left(0, \frac{1}{n_{ijk}-3}\right)$, under the null hypothesis. For $k = 1, 2$, if the $p - value \geq 0.001$ in both groups, then we filter out the pair $(i,j)$ from rest of the calculations.

For each pair $(i,j)$ that was not filtered out in the previous step we tested $H_{0ij}: \theta_{ij1} = \theta_{ij2}$, against the alternative $H_{aij}: \theta_{ij1} \neq \theta_{ij2}$ by invoking $\hat{\theta}_{ij1} - \hat{\theta}_{ij2} \sim^{asymp} N\left(0, \frac{1}{n_{ij1}-3} + \frac{1}{n_{ij2}-3}\right)$, under the null hypothesis. Let $p_{ij}$ denote the corresponding asymptotic 2-sided $p$-value. $p$-values were corrected for multiple hypothesis testing using the BH procedure. To declare if the pair of features $(i,j)$ is differentially correlated, we use the following stringent criteria: (a) the raw $p$-value $p_{ij} < 0.01$, (b)

differences in the estimated correlations (in absolute value) exceed 0.3, i.e., $|\hat{\rho}_{ij1} - \hat{\rho}_{ij2}| > 0.3$, and (c) $q_{ij} < 0.1$. Using these differential correlations, we obtained network plots provided in Fig. 3 (for gut microbiome) and Supplementary Fig. 7 (for oral microbiome) using the NetCoMi[74] v.1.1.0 package in R. In the networks, pairwise correlations with $q < 0.05$ were considered as significant and presented by a solid line, while those with $0.05 \le q < 0.1$ were considered as suggestive and presented by a dashed line.

**Trend analysis of correlations between features over the sexual activity groups.** Withing each sexual activity groups, Spearman correlations among species as well as between species and gene functions (GO terms) and metabolites were computed using complete data. The difference between the previous section and this analysis is that we now have four ordered groups instead of two groups. Hence, we use the same notations as in the previous section, but $k = 1, 2, 3, 4$ representing the four sexual activity groups $G_1, G_2, G_3$, and $G_4$. For each pair of features $(i, j)$ from modality $m$, that are differential correlated between Pre-HIV and Non-HIV at $q < 0.10$ by the analysis noted above, we test if the correlation coefficients have an increasing or decreasing trend over the sexual activity groups. Thus, we test:

$$H_{0ij}: \quad \theta_{ij1} = \theta_{ij2} = \theta_{ij3} = \theta_{ij4}, \quad \text{against the alternative}$$
$$H_{aij}: \{\theta_{ij1} \le \theta_{ij2} \le \theta_{ij3} \le \theta_{ij4}\} \cup \{\theta_{ij1} \ge \theta_{ij2} \ge \theta_{ij3} \ge \theta_{ij4}\} - H_{0ij}.$$

Following the general methodology of ref. 75, we estimate $(\theta_{ij1}, \theta_{ij2}, \theta_{ij3}, \theta_{ij4})'$ under each of the alternative hypotheses $\{\theta_{ij1} \le \theta_{ij2} \le \theta_{ij3} \le \theta_{ij4}\}$ and $\{\theta_{ij1} \ge \theta_{ij2} \ge \theta_{ij3} \ge \theta_{ij4}\}$ by applying Pool Adjacent Violator Algorithm (PAVA) on the unconstrained estimators $\hat{\theta}_{ij1}, \hat{\theta}_{ij2}, \hat{\theta}_{ij3}$, and $\hat{\theta}_{ij4}$. We use inverse of the variances as the weights for the $k^{th}$ group, i.e., $w_k(i,j) = (n_{ijk} - 3)$. Denote the PAVA estimator under $\{\theta_{ij1} \le \theta_{ij2} \le \theta_{ij3} \le \theta_{ij4}\}$ by $\{\tilde{\theta}_{ij1} \le \tilde{\theta}_{ij2} \le \tilde{\theta}_{ij3} \le \tilde{\theta}_{ij4}\}$ and the PAVA estimator under $\{\theta_{ij1} \ge \theta_{ij2} \ge \theta_{ij3} \ge \theta_{ij4}\}$ by $\{\check{\theta}_{ij1} \ge \check{\theta}_{ij2} \ge \check{\theta}_{ij3} \ge \check{\theta}_{ij4}\}$. Then, following Peddada et al. (2003)[75] define Williams' test as follows:

$$Z_1(i,j) = \frac{\tilde{\theta}_{ij4} - \tilde{\theta}_{ij1}}{\sqrt{\frac{1}{n_{ij4}-3} + \frac{1}{n_{ij1}-3}}}, \qquad Z_2(i,j) = \frac{\hat{\theta}_{ij1} - \hat{\theta}_{ij4}}{\sqrt{\frac{1}{n_{ij4}-3} + \frac{1}{n_{ij1}-3}}} \qquad \text{and} \qquad Z(i,j) = \max(Z_1(i,j), Z_2(i,j)).$$

The $p$-value for the test statistic $Z(i,j)$ is simulated as follows.

Step 1: Generate $U_i \sim^{iid} N(0,1), i = 1, 2, 3, 4$.

Step 2: With weights $w_{ijk} = n_{ijk} - 3$, apply PAVA on $\left\{\frac{U_1}{\sqrt{n_{ij1}-3}}, \frac{U_2}{\sqrt{n_{ij3}-3}}, \frac{U_3}{\sqrt{n_{ij3}-3}}, \frac{U_4}{\sqrt{n_{ij4}-3}}\right\}$ under the constraint $\{\theta_{ij1} \le \theta_{ij2} \le \theta_{ij3} \le \theta_{ij4}\}$. Denote the resulting values by $(\tilde{V}_{ij1}, \tilde{V}_{ij2}, \tilde{V}_{ij3}, \tilde{V}_{ij4})'$.

Step 3: Repeat Step2 under the constraint $\{\theta_{ij1} \ge \theta_{ij2} \ge \theta_{ij3} \ge \theta_{ij4}\}$. Denote the resulting values by $(\check{V}_{ij1}, \check{V}_{ij2}, \check{V}_{ij3}, \check{V}_{ij4})'$.

Step 4: We define
$$W_1(i,j) = \frac{\tilde{V}_{ij4} - \tilde{V}_{ij1}}{\sqrt{\frac{1}{n_{ij4}-3} + \frac{1}{n_{ij1}-3}}}, \qquad W_2(i,j) = \frac{\check{V}_{ij1} - \check{V}_{ij4}}{\sqrt{\frac{1}{n_{ij4}-3} + \frac{1}{n_{ij1}-3}}} \qquad \text{and} \qquad W(i,j) = \max(W_1(i,j), W_2(i,j)).$$

Step 5: Step 1 through Step 4 are repeated $N = 1,000,000$ times, yielding the distribution of $W(i,j)$. Then the $p$-value of $Z(i,j)$ is the proportion of times $W(i,j)$ exceeds $Z(i,j)$ out of 1,000,000 simulated runs.

Step 6: Perform multiple testing correction on each $p$-value derived in Step 5 using the BH procedure. Obtain the resulting $q$-value.

Step 7: If the null hypothesis is rejected at the desired $q$-value in Step 6, then declare that the correlation between pair $(i, j)$ is increasing with number of sexual partners if $Z(i,j) = Z_1(i,j)$. If the null is rejected and $Z(i,j) = Z_2(i,j)$, then declare the correlation between pair $(i, j)$ is decreasing with number of sexual partners.

**DISruption in COrrelations (DISCO).** Differences in the environment or conditions between two ecosystems may potentially result in differential interactions among some features between the two ecosystems. For example, as seen in this article, some of the microbial species (as well as various metabolites and functional GO categories) are differentially correlated between Pre-HIV and Non-HIV participants. Consequently, we hypothesize that these differential interactions may be associated with differences in the underlying biology of the two ecosystems, and hence, potentially a state of dysbiosis in an ecosystem (e.g., Pre-HIV) relative to the "reference" ecosystem (e.g., Non-HIV). We therefore describe a taxon to be disrupted in correlations between two ecosystems if it is differentially correlated with other features between the two ecosystems. Accordingly, for a given microbial species, in this article, we introduce a measure of DISruption in COrrelations (DISCO) as follows. Larger the DISCO score, more disrupted the species is.

Suppose $\boldsymbol{X}_k = (X_{1k}, X_{2k}, \ldots, X_{lk})'$ is a $l \times 1$ vector of microbial species in the $k^{th}$ group, $k = 1, 2$. For example, $k = 1$ for Non-HIV and $k = 2$ for Pre-HIV groups. Suppose there are $M$ different modalities of data, such as gut microbial species, gut metabolites, gut GO terms, and plasma metabolites and so on. Suppose there are $f_m$ features within the $m^{th}$ modality of data, $m = 1, 2, \ldots, M$. Let the normalized data for the $m^{th}$ modality of data be denoted by $\boldsymbol{Y}_{mk} = (Y_{m1k}, Y_{m2k}, \ldots, Y_{mf_mk})'$. As in rest of the paper, by normalization we mean bias corrected for compositionality, and adjusted for various covariates of interest. Let $\rho_{mijk}$ denote the population Spearman correlation coefficient between $X_{ik}$ and $Y_{mjk}$, and let $\hat{\rho}_{mijk}$ denote the corresponding sample estimator. Thus, for a given species $i$, let $\boldsymbol{\rho}_{mik} = (\rho_{mi1k}, \rho_{mi2k}, \ldots, \rho_{mif_mk})'$ denote the vector of its true unknown population Spearman correlation coefficients with all features in $m$ for the $k^{th}$ group and $\hat{\boldsymbol{\rho}}_{mik} = (\hat{\rho}_{mi1k}, \hat{\rho}_{mi2k}, \ldots, \hat{\rho}_{mif_mk})'$ denote the vector of estimated correlation coefficients. We denote the corresponding Fisher's Z-transformed correlation coefficients by $\boldsymbol{\theta}_{mik} = (\theta_{mi1k}, \theta_{mi2k}, \ldots, \theta_{mif_mk})'$ and $\hat{\boldsymbol{\theta}}_{mik} = (\hat{\theta}_{mi1k}, \hat{\theta}_{mi2k}, \ldots, \hat{\theta}_{mif_mk})'$, where $\theta_{mijk} = \tanh^{-1}\rho_{mijk}$ and $\hat{\theta}_{mijk} = \tanh^{-1}\hat{\rho}_{mijk} - \frac{\hat{\rho}_{mijk}}{2(n_{mijk}-1)}$, the (asymptotic) bias corrected estimator of $\theta_{mijk}$, and $n_{mijk}$ is the number of samples with non-missing values for the species $i$ and $j^{th}$ feature in the $m^{th}$ modality in the $k^{th}$ group. To deal with sparsity, as done in rest of the paper, we perform hard thresholding to filter out small correlation coefficients. Specifically, for each feature $j$ within the $m^{th}$ modality, for $k = 1, 2$, we test $H_0 : \theta_{mijk} = 0$ vs. $H_a : \theta_{mijk} \ne 0$. Let the number of pairs that are not filtered by the above process be denoted by $f_{mi}$ ($0 \le f_{mi} \le f_m$).

For the species $i$ and modality $m$ with $0 < f_{mi}$, for each pair $(i, j)$ that was not filtered, we test $H_{0mij}: \theta_{mij1} = \theta_{mij2}$, against the alternative $H_{amij}: \theta_{mij1} \ne \theta_{mij2}$ as done in the previous section. Let $p_{mij}$ denote the corresponding asymptotic 2-sided $p$-value. Although under $H_{0mij}, j = 1, 2, \ldots f_m$, the $p$-values $p_{mij}$ are uniformly distributed, they are not independent, and hence standard Fisher's approach to combining $p$-values is not valid. For each taxon $i$, we therefore adopt Cauchy $p$-value combination approach for combining correlated $p$-values. This strategy is widely used for combining correlated $p$-values when analyzing genomic and other high-dimensional data[76]. For species $i$ and modality $m$ (with $0 < f_{mi}$), we define its DISCO score using the Cauchy average of $p$-values: $D_{mi} = \frac{1}{f_{mi}} \sum_{j=1}^{f_{mi}} \mathrm{Tan}\left((0.5 - p_{mij})\pi\right)$. Thus, if a taxon is significantly differentially correlated with several features in the $m^{th}$ modality with small $p$-values then $D_{mi}$ is likely to be large. Thus, according to our definition, larger the value of $D_{mi}$ the more disrupted is species $i$ with respect to modality $m$. As done in the literature[76], we evaluate the statistical significance of $D_{mi}$ by computing $p$-value of $D_{mi}$ using the standard Cauchy (0,1) distribution. Denote the $p$-value by $p_{D_{mi}}$ and let $q_{D_{mi}}$ denotes its $q$-value. For a given modality $m$, we declare the DISCO score for species $i$ to be significant if $q_{D_{mi}} < 0.1$, and $p_{D_{mi}} < 0.01$, and $f_{mi} > .2\max_j f_{mj}$. To be conservative, for each species $i$, we require the additional condition $f_{mi} > .2\max_j f_{mj}$ so that species that are disrupted in their correlations with very few features in modality $m$ are filtered out. We validated the performance of

DISCO by applying it to four external datasets as well as null data and non-null data generated from the gut species abundances using the MACS cohort. The details can be found in Supplementary Information.

Although we described DISCO using Spearman correlations, since the DISCO score is entirely based on the $p$-values corresponding to an association measure, DISCO is general enough that one can use any measure of association or dependence, such as partial correlations, distance correlations, Chatterjee correlation and others. Thus, we are offering a general framework to describe disruption in correlations.

**Significance and error-rate control.** In all the above analyses, to control the false discovery rate, we implemented the BH procedure at the feature level and we reported features with $q < 0.1$, considering $q < 0.05$ as significant and $0.05 \leq q < 0.1$ as suggestive. Given the possibility of inflation in the BH procedure due to the correlated structure of high-dimensional data and its potential non-linear relationship with $p$-values, we further restricted the results to those with $p < 0.01$. Additional restrictions were imposed based on the type of analysis. For example, in differential correlation analysis, all significant/suggestive differential correlated features were required to have an absolute correlation difference of at least 0.3 between the two groups. Per-feature, the $p$ and $q$ values are available in the supplementary Data.

All the statistical analyses were performed in RStudio 2023.09.0 using R v. 4.4.1.

### Reporting summary
Further information on research design is available in the Nature Portfolio Reporting Summary linked to this article.

## Data availability
Access to individual-level data from the MACS/WIHS Combined Cohort Study Data (MWCCS) may be obtained upon review and approval of a MWCCS concept sheet. Links and instructions for online concept sheet submission are on the study website (http://mwccs.org/). Once the concept sheet is approved, please contact Dr. Yue Chen (cheny@pitt.edu) to obtain the metadata. The metagenomic data generated in this study are available under NCBI BioProject ID PRJNA1303200. The metabolomic data generated in this study are available at https://www.metabolomicsworkbench.org/[77] under project PR002653 [https://doi.org/10.21228/M8QN9C] with studies ST004208, ST004209, and ST004210 for plasma, oral, and stool metabolomics, respectively. The raw data underlying figures are available in Source Data. Source data are provided with this paper.

## Code availability
Codes can be accessed from https://github.com/FarnazFouladi/Microbiome_Metabolomics_HIV.git [https://doi.org/10.5281/zenodo.17179699].

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

## Acknowledgements
Oral wash, stool, and blood plasma specimens and epidemiologic data in this manuscript were originally collected as part of the Multicenter AIDS Cohort Study (MACS), now part of the MACS/WIHS Combined Cohort Study (MWCCS), and provided by the MWCCS Data and Coordination Center (DACC; U01-HL146193). The authors thank especially the original, 1984–1985 participants of the Multicenter AIDS Cohort Study, without whom this study would not be possible. DNA for the present study was extracted by the Rinaldo lab at the University of Pittsburgh MWCCS Clinical Research Site U01-HL146208 (MPIs Charles Rinaldo and Jeremy Martinson). We thank Arlene Bullotta, Kathy Kulka, Kathy Hartle, and Dr. Aiymkul Ashimkhanova from the Rinaldo laboratory for extracting DNA from the stool and oral wash samples. The works of S.D.P., F.F., and S.B. are supported [in part] by the National Institute of Environmental Health Sciences (NIEHS) Intramural Research Program ZIA ES103400-01 and ZIA ES103389-01. The NIH intramural funding from the Office of AIDS Research (OAR) to SDP supported sample preparation, DNA extraction, and the generation of microbiome, metabolomics, and SCFA data. The work of JA was supported [in part] by the National Institute of Environmental Health Sciences (NIEHS) Intramural Research Program ZIC ES103363. We thank the NIEHS intramural reviewers, Dr. Bill Copeland (Genome Integrity and Structural Biology Laboratory) and Dr. Mikyeong Lee (Epidemiology Branch) for numerous constructive comments that improved the presentation of the paper. We thank Dr. John McCulloch, Microbiome and Genetics core, Laboratory of Integrative Cancer Immunology, NCI, NIH. We also thank Dr. Geoff Muller, Director, Nuclear Magnetic Resonance Research Core Facility, Genome Integrity and Structural Biology, NIH, for analysis of a pilot oral wash data. Metabolomics Workbench is supported by NIH grant U2C-DK119886 and OT2-OD030544 grants. The contents of this publication are solely the responsibility of the authors and do not represent the official views of the National Institutes of Health (NIH) or the OAR. The authors gratefully acknowledge the dedication and contributions of the study participants and staff at the MACS/MWCCS clinical and data management sites.

## Author contributions
F.F.: methods; data analysis; interpretation of results; manuscript preparation; editing. Y.C.: study design; DNA extraction and sample preparation; editing. S.B.: methods; data analysis; editing. A.K.J.: methods; editing. V.T.D.: editing. F.J.P.: editing. J.B.M.: editing. K.W.C.: editing. J.S.: editing. J.M.: editing. C.R.R.: study design; DNA extraction and sample preparation; editing. S.D.P.: study conception and aims; study design; methods; interpretation of results; manuscript preparation; editing.

## Funding

## Competing interests
The authors declare no competing interests.

## Additional information

[1]Biostatistics and Computational Biology Branch, Division of Intramural Research, National Institute of Environmental Health Sciences, National Institutes of Health, Research Triangle Park, Durham, NC, USA. [2]Division of Infectious Diseases, Department of Medicine, University of Pittsburgh School of Medicine, Pittsburgh, PA, USA. [3]Immunity, Inflammation, and Disease Laboratory, Division of Intramural Research, National Institute of Environmental Health Sciences, National Institutes of Health, Research Triangle Park, Durham, NC, USA. [4]Division of Infectious Diseases, Feinberg School of Medicine, Northwestern University, Chicago, IL, USA. [5]Department of Molecular Microbiology and Immunology, Johns Hopkins Bloomberg School of Public Health, Johns Hopkins University, Baltimore, MD, USA. [6]Division of Infectious Diseases, Department of Medicine, David Geffen School of Medicine at University of California, Los Angeles, Los Angeles, CA, USA. [7]Department of Epidemiology, Bloomberg School of Public Health, Johns Hopkins University, Baltimore, MD, USA. [8]Department of Infectious Diseases and Microbiology, School of Public Health, University of Pittsburgh, Pittsburgh, PA, USA. ✉e-mail: peddada@nih.gov

