## [Transparent Peer Review file · Nature Communications]

A taxon-specific measurement of disruption in a multi-modal study of microbiomes and metabolomes reveals system-wide dysbiosis preceding HIV-1 infection

Corresponding Author: Dr Shyamal Peddada

Version 0:

Reviewer comments:

Reviewer #1

(Remarks to the Author)

This study from Fouladi et al investigates the interactions between the microbiome and metabolome in the period immediately prior to HIV acquisition using samples from the Multicenter AIDS Cohort Study. The findings are interesting and largely in agreement with previously published studies, with some novel associations identified here.

1. There is not much data given on the study population and how the Pre-HIV group compares to the Non-HIV group. Even when looking in the referenced paper, there is still only basic characteristics such as age, study site location, and timing of sample listed in Table 1. What is the ethnic/racial makeup of the group? How does sexual activity compare? Substance use? Smoking? All of these can affect the microbiome and/or inflammatory state and could be potentially important confounders.

2. Figure 3 does not add much to the study and would be better in supplementary materials. The relevant portion of this figure is shown in Figure 4 where the actual interaction can actually be visualized.

3. The DYSCO score, while an interesting idea, does not feel fully developed. What would the authors expect the typical variation in DYSCO scores between population groups to be? For example, if random groups were created within the Non-HIV group (which is deemed "normal healthy ecology" in this analysis) are there still differences detected? Just because a species has different correlations with metabolites, etc between groups, does not necessarily mean the overall phenotype has changed and it constitutes dysbiosis. It's possible it does, but I don't think this can be assumed. Perhaps viewing this analysis as identifying 'altered interactions' or similar is more appropriate, but determining that all these species are 'dysbiotic' based on this analysis seems like a bit of an over interpretation.

4. It is not clear how the DYSCO score is corrected for multiple comparisons. Testing the differences between groups for the correlation of a given species with all the other data modalities (and doing that for many species) results in an extraordinary number of comparisons.

(Remarks on code availability)

Reviewer #2

(Remarks to the Author)

Key Results

In this study, the authors analyze differences in the microbiome (intestinal and oral) and metabolome (fecal and plasma) in samples collected prior to HIV-1 infection in a prospective cohort of men who have sex with men (MSM). The authors identify a generalized dysbiosis prior to HIV-1 infection, highlighting changes in bacterial composition, metabolic functions such as purine and amino acid metabolism, and an increased presence of metabolites indicative of oxidative stress. Furthermore, they introduce a novel method (the DYSCO index) to analyze ecological differences in the microbiome.

Validity

Although the study presents a technically robust methodology in terms of data generation (shotgun metagenomics and untargeted metabolomic analysis), the interpretation and presentation of results reveal important issues:

- a. The interpretation is frequently speculative, especially when the authors suggest direct biological mechanisms based solely on statistical correlations. This detracts from the validity of the conclusions, as no functional or experimental data are provided to support causality.
- b. Another key issue is the repeated use of marginally significant results ($p < 0.01$) that do not reach significance after correction for multiple testing ($q > 0.1$). This leads to potentially exaggerated conclusions, particularly since the article does not present any type of experimental validation or validation with other datasets.

These limitations do not necessarily preclude publication, but they require significant corrections before the findings can be considered robust.

Significance

The work could have significant relevance by suggesting that dysbiosis precedes HIV infection, thereby opening potential preventive avenues based on the microbiome. However, the omission of recent literature that attributes microbiological differences to sexual practices rather than HIV (10.1016/j.ebiom.2016.01.032), or the failure to consider sexual behavior as a covariate in differential abundance analyses—even when previous work by the authors has addressed this (e.g., “Sexual behavior is linked to changes in gut microbiome and systemic inflammation that lead to HIV-1 infection in men who have sex with men, <https://doi.org/10.1038/s42003-024-06816-z>”)—considerably limits the originality and relevance attributed to the finding. It is important to incorporate and explicitly discuss these references and use sexual behavior variables as confounders in all analysis.

Data and Methodology

The quality of the data is high, and the methodology is adequate, but there are several deficiencies that need to be addressed:

- a. **Metabolomics:** The description of the methods used for generating the metabolomic data is extremely brief. Outsourcing the analysis does not justify this lack of detail. This section should be expanded to ensure reproducibility.
- b. **Cohort Description:** Referring the reader to a previous publication to consult basic demographic characteristics is inconvenient and impractical. I strongly recommend including this table in the supplementary material of the current manuscript.
- c. **GitHub Repository:** The indicated repository is not accessible (incorrect link, private, or restricted), significantly limiting reproducibility. It is essential to ensure public accessibility prior to publication.

Analytical Approach

The overall analytical strategy is adequate (ANCOM-BC2, multi-omics integration with DIABLO), but some issues are identified:

- a. **Confounding Variables:** It is highly concerning that the authors have not systematically adjusted for sexual behavior / preference, especially considering that the results themselves (section on DYSCO score and <https://doi.org/10.1038/s42003-024-06816-z>) clearly show the importance of this factor. This omission could partially invalidate the main conclusions.
- b. **Excessive Use of Marginal Results:** It is recommended to clearly present results adjusted for multiple testing (q-values) rather than relying excessively on unadjusted results (p-values). This is particularly critical in this work, as it does not include any experimental validation or validation using independent cohorts.
- c. **DYSCO Index:** Although innovative, it is highly susceptible to uncontrolled confounders. This vulnerability should be clearly emphasized in the interpretation, even though the method does allow for correction if the confounders are known.

Suggested Improvements

To substantially strengthen the work before publication, I recommend:

1. Conducting additional analyses with ANCOM-BC2 that explicitly adjust for variables related to sexual behavior (number of partners, number of sexual encounters, etc.).
2. Significantly improving the methodological description of the metabolomic analysis.
3. Ensuring open accessibility of the GitHub repository.
4. Reducing or simplifying the number of network graphs, using alternative, clearer formats (simplified diagrams, heatmaps, dotplots, tables).
5. Including a descriptive table of the cohort in the supplementary material.
6. Carefully correcting errors in taxonomic names throughout the manuscript (e.g., "Holdemonella" instead of "Holdemanella" or "Gemming" instead of "Gemmiger").
7. Presenting only those results that reach statistical significance after correction for multiple testing or clearly emphasizing this limitation and adjusting the presentation of results and discussion with cautious language.

I think that all these suggestions can be addressed without substantially changing the focus of the work, but they will significantly improve its scientific quality.

Clarity and Context

Overall clarity is acceptable, but the discussion contains speculative interpretations based mainly on correlations, which require significant moderation. Additionally, there is a lack of explicit contextualization regarding previous literature that could limit the interpretation of the results (as mentioned earlier).

We hope these comments help improve the manuscript.

(Remarks on code availability)

The indicated GitHub repository is not accessible (incorrect link, private, or restricted), significantly limiting reproducibility.

Reviewer #3

(Remarks to the Author)

(Remarks on code availability)

GitHub Repo not accessible

Version 1:

Reviewer comments:

Reviewer #2

(Remarks to the Author)

The updated manuscript demonstrates significant improvements in clarity and depth, successfully resolving many earlier issues. The discussion now reflects the data more accurately, and the methods are described with increased detail, bringing the work close to being ready for publication. Nevertheless, important challenges persist around statistical transparency, reproducibility, and how results are interpreted across various multi-omic layers. These points are detailed below, accompanied by targeted recommendations to assist the authors in addressing them effectively.

1. Statistical thresholds — clarify the discovery policy and, if retained, relabel suggestive findings.

In the current draft, “significance” is defined as $p < 0.01$ and $q < 0.10$, with asterisks marking values where $q \leq 0.10$. For exploratory studies without an independent validation cohort, a threshold of $q = 0.10$ can be considered lenient, and the use of star notation may not clearly distinguish between statistically significant and suggestive results. It is recommended to either provide a concise rationale for using $q \leq 0.10$ (such as balancing discovery power across multiple features or modalities) or adjust the threshold to $q \leq 0.05$ for results classified as “significant.” If maintaining $q \leq 0.10$, relabel results where $0.05 < q \leq 0.10$ as “suggestive” in both text and figures, reserve “significant” for $q \leq 0.05$, and modify the asterisk legend accordingly. Additionally, include a paragraph in the Methods section titled “Significance and error-rate control,” specifying the primary FDR procedure (feature-level BH), detailing the labeling policy, and indicating where per-feature p/q values are available in the supplement.

2. Sample-size consistency

There is inconsistency in reporting the Non-HIV sample count, as it appears as 149 in some parts and 152 in others. Please make sure to use one correct number consistently across the entire manuscript.

3. Minor erratum — correct the implausible reference year

In the manuscript, the citation “Armstrong et al., 2028” appears, which is likely a typographical error. Kindly ensure this is corrected throughout the document, including in figure legends, references, and any other relevant sections.

4. I’ve been asked to comment on the analysis performed in response to Reviewer #1’s Point 3: I have a fundamental issue with the central assumption of the analysis. Specifically, I’m not sure that defining “dysbiosis” as a random Cauchy variable is correct. Typically, dysbiosis is computed from ecological distances or log-ratio transforms, not by assuming a particular parametric distribution for the underlying abundances or the index itself. Additionally, dysbiosis is an ecological term defined for the entire community, so it doesn’t make sense to me to state that “particular species were dysbiotic,” as suggested with the *Prevotella* example in the rebuttal.

(Remarks on code availability)

The results presented in the paper are reproducible, and the code serves as a useful resource for the community. The code includes a README file with instructions for installation and execution. It is possible to install and run the code.

Reviewer #3

(Remarks to the Author)

(Remarks on code availability)

Response to reviewers' comments

We thank the reviewers and the editors for their very helpful comments and suggestions which we believe substantially improved the manuscript. We addressed all their comments in the revision. In the following we provide item by item response to the comments we received from each reviewer. Their comments are in italics and our responses follow them in plain text.

Reviewer #1 (Remarks to the Author):

This study from Fouladi et al investigates the interactions between the microbiome and metabolome in the period immediately prior to HIV acquisition using samples from the Multicenter AIDS Cohort Study. The findings are interesting and largely in agreement with previously published studies, with some novel associations identified here.

1. There is not much data given on the study population and how the Pre-HIV group compares to the Non-HIV group. Even when looking in the referenced paper, there is still only basic characteristics such as age, study site location, and timing of sample listed in Table 1. What is the ethnic/racial makeup of the group? How does sexual activity compare? Substance use? Smoking? All of these can affect the microbiome and/or inflammatory state and could be potentially important confounders.

RESPONSE: We thank the reviewer's comment. This information is now included in Table 1. We also noticed that there were 3 individuals in the Non-HIV group who had CCR5-delta32 homozygous genotype and were hence naturally protected against HIV-1 infection. Hence, we removed them from our analysis, and this resulted in 152 subjects in the Non-HIV group.

2. Figure 3 does not add much to the study and would be better in supplementary materials. The relevant portion of this figure is shown in Figure 4 where the actual interaction can actually be visualized.

RESPONSE: We agree with the reviewer's comment that Figure 3 is not informative, and therefore, we removed it from the figures and only kept Figure 4 (named as Figure 3 in the revised manuscript). Additionally, we simplified the networks to show the differences in correlations between Pre-HIV and Non-HIV rather than showing the correlations in Pre-HIV and Non-HIV with separate networks.

3. The DYSCO score, while an interesting idea, does not feel fully developed. What would the authors expect the typical variation in DYSCO scores between population groups to be? For example, if random groups were created within the Non-HIV group (which is deemed "normal healthy ecology" in this analysis) are there still differences detected? Just because a species has different correlations with metabolites, etc. between groups, does not necessarily mean the overall phenotype has changed and it constitutes dysbiosis. It's possible it does, but I don't think this can be assumed. Perhaps viewing this analysis as identifying 'altered interactions' or similar is more appropriate, but determining that all these species are 'dysbiotic' based on this analysis seems like a bit of an over interpretation.

RESPONSE: We thank the reviewer for this comment. In view of the reviewer's comment, we made the following changes:

- (a) We modified the definition of our measure of dysbiosis which is now defined as a Cauchy random variable. The larger the value the more likely the taxon is dysbiotic with respect to a given modality of data. The associated p-value can be derived from the standard Cauchy (0,1) distribution. Results presented in the paper are after performing corrections for multiple testing using the Benjamini-Hochberg procedure-based q-value. Please refer to the Methods section for details. As recommended by the reviewer, we performed a validation study as follows.

Bootstrap – Null data (A): We combined the Pre-HIV group ($n_1 = 86$) and Non-HIV group ($n_2 = 149$) into a single population and then randomly split the data into two groups of sizes $n_1 = 86$ and $n_2 = 149$ with replacement. We then computed the DYSCO score for each taxon between two randomly created groups. We repeated this process 1000 times. Since the Cauchy random variable has no mean, we computed the median DYSCO score over the 1000 simulations for each taxon and created the boxplot (left hand side of the below figure).

Bootstrap – Non-null– Real data (B): We drew a random sample with replacement of size 86 from the Pre-HIV group ($n_1 = 86$) and a random sample with replacement of size 149 from the Non-HIV group ($n_2 = 149$). As in (A), we then computed the DYSCO score for each taxon between two randomly created groups. We repeated this process 1000 times. Thus, we bootstrapped samples within the real data. Again, since the Cauchy random variable has no mean, we computed the median DYSCO score over the 1000 simulations for each taxon and created the boxplot (right hand side of the below figure).

Comparison of Median Scores between Null and Real Data

Since data in (A) are created by mixing the samples from the two groups, the resulting DYSCO score for each taxon can be viewed as the score under the Null, whereas since the DYSCO score for data in (B) are based on two different phenotypes, it may be regarded as score under the non-null. As expected, the distribution of median DYSCO scores of taxa derived from real data is stochastically larger than that derived from null data. This validation study has been included in the revised manuscript in the Supplementary text, page 24.

- (b) In response to the comment whether the phenotype is dysbiotic, we do not claim that changes in the correlations of a taxon with other features cause dysbiosis. Instead, we hypothesize that these differential interactions may be associated with differences in the underlying biology of the two ecosystems, and hence, potentially a state of dysbiosis in an ecosystem (e.g. Pre-HIV) relative to the “reference” ecosystem (e.g. Non-HIV). We therefore describe a feature to be dysbiotic between two ecosystems if it is differentially correlated with other features between the two ecosystems. Accordingly, for a given microbial species, we introduce DYSCO a measure of DYSbiosis based on COrrrelations. Larger the DYSCO score, more dysbiotic the species is.

Several methods for quantifying the microbial dysbiosis have been previously introduced in the literature and summarized in Wei, S., et al (2021). Many of these methods compare the taxonomic profiles in the sample with a population-based reference. For example, Gut Microbiome Health Index (GMHI) (Gupta et al., 2020) or in Gut Microbiome Wellness Index 2 (GMWI2) (Chang et al., 2024) is an index that represents the dysbiotic level of a sample based on the distribution of healthy and disease-related species, which are defined from an integrated dataset of thousands of publicly available metagenomes from various studies including different types of diseases. In some other

methods dysbiosis scores are measured based on differentially abundant taxa (Gevers et al., 2014) (e.g., the difference/ratio between abundance of taxa enriched in a disease group and those enriched in a healthy group), or based on alpha and beta diversity measurements (Lloyd-Price et al., 2019, and Le Chatelier et al, 2013). The dysbiosis score introduced in this paper is different with the previous ones in many regards. First, our method is species-specific rather than sample-specific, i.e., instead of estimating a dysbiosis score for a sample we estimate a dysbiosis score for a species. Second, our method is based on the changes in the interaction of microbiome with other biological components in a group of interest relative to a reference group. Thus, we say a taxon is dysbiotic if its interactions with other features, such as the metabolites and GO terms, changes between two phenotypes (or environments) and we provide a measure of this dysbiosis. The measure highlights species with perturbed interactions, which could potentially shed light on the mechanisms by which a species could contribute to a specific disease or be a consequence of the disease. By knowing if a taxon is dysbiotic between two conditions in terms of its interactions with other parameters we gain insights about its importance to the overall ecosystem. We have highlighted the above discussion on page 9 in the paper.

- (c) In response to the editor's suggestion, we also wish to highlight here that we applied DYSCO on four other HIV data sets and made several interesting discoveries. For example, we found that across studies, *Prevotella* species were dysbiotic and had a significant DYSCO score (Supplementary Figure 9 and 10). This finding is reassuring because proinflammatory properties of *Prevotella* spp. are well-known, especially in the context of HIV-1. Details are provided in the Supplementary text on pages 27-29.

4. *It is not clear how the DYSCO score is corrected for multiple comparisons. Testing the differences between groups for the correlation of a given species with all the other data modalities (and doing that for many species) results in an extraordinary number of comparisons.*

RESPONSE: We thank the reviewer for this comment. As noted above, for each modality, for each taxon we have now implemented multiple testing corrections (based in the BH procedure) to our DYSCO score and report taxa with $p < 0.01$ and $q < 0.1$.

Reviewer #2 (Remarks to the Author):

Key Results

In this study, the authors analyze differences in the microbiome (intestinal and oral) and metabolome (fecal and plasma) in samples collected prior to HIV-1 infection in a prospective cohort of men who have sex with men (MSM). The authors identify a generalized dysbiosis prior to HIV-1 infection, highlighting changes in bacterial composition, metabolic functions such as purine and amino acid metabolism, and an increased presence of metabolites indicative of oxidative stress. Furthermore, they introduce a novel method (the DYSCO index) to analyze ecological differences in the microbiome.

Validity

Although the study presents a technically robust methodology in terms of data generation (shotgun metagenomics and untargeted metabolomic analysis), the interpretation and presentation of results reveal important issues:

a. The interpretation is frequently speculative, especially when the authors suggest direct biological mechanisms based solely on statistical correlations. This detracts from the validity of the conclusions, as no functional or experimental data are provided to support causality.

RESPONSE: We thank the reviewer for this comment. It is true that we are interpreting the results with the help of published literature, and additional experimental data are required to verify the results and to draw conclusions about causality. The focus of this paper is to understand and explain system wide changes that may be occurring before the onset of HIV using 10 different modalities of data. Although the GO terms and metabolomics provide some functional information described in this paper, ideally one may want to conduct wet-lab in-vitro or in-vivo studies, which are beyond the scope of our paper. In the discussion section we include a comment on page 9 to acknowledge this limitation: “the present study is an association study and does not provide evidence for causality. Although we have provided mechanistic insights with the analysis of GO terms and metabolomes, *in vitro* and/or *in vivo* studies will be required to validate our findings”.

Despite this limitation, we investigated the application of DYSCO score to four external datasets as suggested by the editor. Our findings using the DYSCO score strengthened/supplemented the findings of the original manuscripts (Please see the Supplementary text on pages 27-29). This provides evidence that the concept of “dysbiosis based on correlation” could be biologically relevant and meaningful.

b. Another key issue is the repeated use of marginally significant results ($p < 0.01$) that do not reach significance after correction for multiple testing ($q > 0.1$). This leads to potentially exaggerated conclusions, particularly since the article does not present any type of experimental validation or validation with other datasets. These limitations do not necessarily preclude publication, but they require significant corrections before the findings can be considered robust.

RESPONSE: We thank the reviewer’s comment about the correction for multiple testing. We have now implemented multiple testing corrections using the BH procedure throughout the

paper. In fact, we used very stringent criteria even beyond the standard BH procedure. Not only did we require the BH based $q < 0.10$ but we imposed further stringency by also requiring the $p < 0.01$, and for differential correlation analysis, we imposed the difference in absolute correlations between a pair of features to be more than 0.3, in addition to $p < 0.01$ and $q < 0.1$ before they are declared significant. In some cases, when there was a biological plausibility, we interpreted the results when the q -value was over 0.1. Such cases are mentioned explicitly in the paper for transparency.

Significance

The work could have significant relevance by suggesting that dysbiosis precedes HIV infection, thereby opening potential preventive avenues based on the microbiome. However, the omission of recent literature that attributes microbiological differences to sexual practices rather than HIV (10.1016/j.ebiom.2016.01.032), or the failure to consider sexual behavior as a covariate in differential abundance analyses—even when previous work by the authors has addressed this (e.g., “Sexual behavior is linked to changes in gut microbiome and systemic inflammation that lead to HIV-1 infection in men who have sex with men, <https://doi.org/10.1038/s42003-024-06816-z”>)—considerably limits the originality and relevance attributed to the finding. It is important to incorporate and explicitly discuss these references and use sexual behavior variables as confounders in all analysis.

RESPONSE: We thank the reviewer for this excellent point. We agree with the reviewer that sexual practices can have a significant impact on the gut microbiome. In our study, the Pre-HIV group had a significantly higher number of sexual partners with whom they had receptive intercourse compared to Non-HIV (Please see Table 1). In other words, the rate of HIV-1 infection increased with the number for sexual partners:

Given that the HIV infection is highly correlated with sexual activity groups (based on the number of sexual partners), statistically it is not appropriate to include both variables in a model for differential abundance testing due to collinearity. Instead, to investigate the effect of sexual activity on the microbiomes and metabolomes, we performed a trend analysis using ANCOM-BC2 and identified microbial and metabolic features that were significantly associated with sexual activity (Supplementary Figures 1 and 5). We also compared the results from the differential abundance testing of microbiomes and metabolomes between Pre-HIV versus Non-

HIV with those from the trend analysis over the four sexually activity groups (Supplementary Figures 2 and 6). Overall, we observed that many observed differences between Pre-HIV and Non-HIV have a decreasing or increasing trend over the sexual activity groups, suggesting that sexual activity could be potentially the driver for changes in the microbiomes and metabolomes prior to HIV-1 infection. However, we also found some differentially abundant microbial features between Pre-HIV versus Non-HIV that were not associated with the number of sexual partners. Examples of such features include the enrichment of *Holdemanella biformis*, *Holdemanella porci*, and *Slackia isoflavoniconvertens* and the reduction of gene functions involved in histidine metabolism and protein secretion by the type VI secretion systems in Pre-HIV. These results suggest that some of the taxa associated with Pre-HIV could be related to other factors rather than sexual activity.

To address the impact of sexual activity on the differential correlation analysis, we implemented a similar strategy as differential abundance testing. We performed a trend analysis using our constrained inference procedures (Peddada et al., 2003) to identify correlations between features that have a decreasing and increasing trend across the four sexual activity groups. This test was reported for all the significant differential correlations reported in Figures 3 and supplementary Figure 7. The correlation coefficients by sexual activity groups (along with adjusted p-values constrained inference test) are visualized in supplementary Figures 3 and 8. Overall, some of the reported significant differential correlations seem to be related to sexual activity groups (such as interactions between gut species between enzyme activities involved in purine and pyrimidine biosynthesis pathways), while some of the differential correlations such as interactions between *Holdemanella spp* and other gut species seem to be unrelated the sexual activity groups.

We thank the reviewer for pointing us to the paper by Noguera-Julian et al. about the link between gut microbiome and sexual preference. We have referred to this paper in our revised manuscript.

Data and Methodology

The quality of the data is high, and the methodology is adequate, but there are several deficiencies that need to be addressed:

a. Metabolomics: The description of the methods used for generating the metabolomic data is extremely brief. Outsourcing the analysis does not justify this lack of detail. This section should be expanded to ensure reproducibility.

RESPONSE: The details are provided in the supplementary methods.

b. Cohort Description: Referring the reader to a previous publication to consult basic demographic characteristics is inconvenient and impractical. I strongly recommend including this table in the supplementary material of the current manuscript.

RESPONSE: We thank the reviewer's comment. The demographic characteristics are now included in Table 1. We also noticed that there were 3 individuals in the Non-HIV group who had CCR5-delta32 homozygous genotype and were hence naturally protected against HIV-1 infection. Hence, we removed them from our analysis, and this resulted in 152 subjects in the Non-HIV group.

c. GitHub Repository: The indicated repository is not accessible (incorrect link, private, or restricted), significantly limiting reproducibility. It is essential to ensure public accessibility prior to publication.

RESPONSE: For the purposes of review of the results presented in this manuscript, we are providing all the processed data along with the codes to the editor and reviewers at https://github.com/FarnazFouladi/Microbiome_Metabolomics_HIV.git. Due to MWCCS policy, we request you to delete all the downloaded MACS cohort data files after the reviewing this paper. These MACS cohort data are available until August 22, 2025. However, if you are unable to download these files by this day, then please communicate through the editor so that we can extend the date suitably.

Once the paper is published, all individual-level data will be available for public use as described in the section “Data and Code Availability”:

“Access to individual-level data from the MACS/WIHS Combined Cohort Study Data (MWCCS) may be obtained upon review and approval of a MWCCS concept sheet. Links and instructions for online concept sheet submission are on the study website (<http://mwccs.org/>). All the codes can be accessed from https://github.com/FarnazFouladi/Microbiome_Metabolomics_HIV.git.”

Analytical Approach

The overall analytical strategy is adequate (ANCOM-BC2, multi-omics integration with DIABLO), but some issues are identified:

a. Confounding Variables: It is highly concerning that the authors have not systematically adjusted for sexual behavior / preference, especially considering that the results themselves (section on DYSCO score and <https://doi.org/10.1038/s42003-024-06816-z>) clearly show the importance of this factor. This omission could partially invalidate the main conclusions.

RESPONSE: Please see the response to the comment on the significance.

b. Excessive Use of Marginal Results: It is recommended to clearly present results adjusted for multiple testing (q-values) rather than relying excessively on unadjusted results (p-values). This is particularly critical in this work, as it does not include any experimental validation or validation using independent cohorts.

RESPONSE: We thank the reviewer’s comment about the correction for multiple testing. As noted above to earlier comment by this reviewer as well as reviewer 1, we have now implemented the BH procedure for multiple testing. In fact, as noted above, we imposed further stringency by also requiring the $p < 0.01$ and the difference in absolute correlations between a pair of features to be at least 0.3 before they are declared significant.

c. DYSCO Index: Although innovative, it is highly susceptible to uncontrolled confounders. This vulnerability should be clearly emphasized in the interpretation, even though the method does allow for correction if the confounders are known.

RESPONSE: We thank the reviewer for pointing to this limitation that uncontrolled confounders may have an effect on DYSCO score. As per the reviewer's comment, we have noted this limitation in the revised manuscript. Please see page 9: "In particular, the effect of diet and other unmeasured confounders on DYSCO is unknown and requires further investigation in future studies".

Suggested Improvements

To substantially strengthen the work before publication, I recommend:

1. *Conducting additional analyses with ANCOM-BC2 that explicitly adjust for variables related to sexual behavior (number of partners, number of sexual encounters, etc.).*

RESPONSE: Please see the response to the comment on the significance.

2. *Significantly improving the methodological description of the metabolomic analysis.*

RESPONSE: The details are provided in the supplementary methods.

3. *Ensuring open accessibility of the GitHub repository.*

RESPONSE: Please see the response to the GitHub Repository.

4. *Reducing or simplifying the number of network graphs, using alternative, clearer formats (simplified diagrams, heatmaps, dotplots, tables).*

RESPONSE: We agree with the reviewer's comment regarding simplifying the network plots. Accordingly, we removed network plots (Figure 3 and Supplementary Figure2) and only kept the sub-networks Figure 4 and Supplementary Figure3 which visualize the significant differential correlations mentioned in the text (These figures are named as Figure 3 and Supplementary Figure7 in the revised manuscript). Additionally, we simplified the networks to show differences in correlations between Pre-HIV and Non-HIV rather than showing the correlations in Pre-HIV and Non-HIV using separate networks.

5. *Including a descriptive table of the cohort in the supplementary material.*

RESPONSE: This information is now included in Table 1.

6. *Carefully correcting errors in taxonomic names throughout the manuscript (e.g., "Holdemonella" instead of "Holdemanella" or "Gemminger" instead of "Gemmiger").*

RESPONSE: We appreciate the reviewer for informing us about the typos. We carefully proofread the revised manuscript and fixed the typos.

7. *Presenting only those results that reach statistical significance after correction for multiple testing or clearly emphasizing this limitation and adjusting the presentation of results and discussion with cautious language.*

RESPONSE: Again, we thank the reviewer for the comment on the multiple hypothesis testing. As mentioned above, we have addressed this issue throughout the paper.

I think that all these suggestions can be addressed without substantially changing the focus of the work, but they will significantly improve its scientific quality.

Response: Thank you so much. We appreciate your comments, and we agree that your suggestions did not change the focus but certainly improved the manuscript substantially.

Clarity and Context

Overall clarity is acceptable, but the discussion contains speculative interpretations based mainly on correlations, which require significant moderation. Additionally, there is a lack of explicit contextualization regarding previous literature that could limit the interpretation of the results (as mentioned earlier).

Response: As noted above, we have addressed your concerns by editing the discussion section suitably.

We hope these comments help improve the manuscript.

Response: Thank you again. Yes, your comments have significantly improved the presentation of our work.

Reviewer #2 (Remarks on code availability):

The indicated GitHub repository is not accessible (incorrect link, private, or restricted), significantly limiting reproducibility.

Response: Please see the response to the GitHub Repository.

Reviewer #3 (Remarks to the Author):

Reviewer #3 (Remarks on code availability):

GitHub Repo not accessible

Response: Please see the response to the GitHub Repository.

Response to Reviewer 2's comments

We thank the reviewer and the editor for their very helpful comments and suggestions which we believe improved the manuscript. We addressed all their comments in the revision. In the following we provide item by item response to the comments we received. Reviewer's comments are in italics and our responses follow them in plain text.

Reviewer #2 (Remarks to the Author):

The updated manuscript demonstrates significant improvements in clarity and depth, successfully resolving many earlier issues. The discussion now reflects the data more accurately, and the methods are described with increased detail, bringing the work close to being ready for publication. Nevertheless, important challenges persist around statistical transparency, reproducibility, and how results are interpreted across various multi-omic layers. These points are detailed below, accompanied by targeted recommendations to assist the authors in addressing them effectively.

1. Statistical thresholds — clarify the discovery policy and, if retained, relabel suggestive findings. In the current draft, “significance” is defined as $p < 0.01$ and $q < 0.10$, with asterisks marking values where $q \leq 0.10$. For exploratory studies without an independent validation cohort, a threshold of $q = 0.10$ can be considered lenient, and the use of star notation may not clearly distinguish between statistically significant and suggestive results. It is recommended to either provide a concise rationale for using $q \leq 0.10$ (such as balancing discovery power across multiple features or modalities) or adjust the threshold to $q \leq 0.05$ for results classified as “significant.” If maintaining $q \leq 0.10$, relabel results where $0.05 < q \leq 0.10$ as “suggestive” in both text and figures, reserve “significant” for $q \leq 0.05$, and modify the asterisk legend accordingly. Additionally, include a paragraph in the Methods section titled “Significance and error-rate control,” specifying the primary FDR procedure (feature-level BH), detailing the labeling policy, and indicating where per-feature p/q values are available in the supplement.

Our response: Thanks for the comments. The reviewer provided us a few options to deal with FDR issue and accordingly we have revised the document using the following option: *“If maintaining $q \leq 0.10$, relabel results where $0.05 < q \leq 0.10$ as “suggestive” in both text and figures, reserve “significant” for $q \leq 0.05$, and modify the asterisk legend accordingly. Additionally, include a paragraph in the Methods section titled “Significance and error-rate control,” specifying the primary FDR procedure (feature-level BH), detailing the labeling policy, and indicating where per-feature p/q values are available in the supplement”.*

2. Sample-size consistency

There is inconsistency in reporting the Non-HIV sample count, as it appears as 149 in some parts and 152 in others. Please make sure to use one correct number consistently across the entire manuscript.

Our response: We were unable to generate data for a few samples due to poor quality. Hence the sample sizes varied slightly across modalities as noted in our Supplementary Table 12 and the method section.

3. Minor erratum — correct the implausible reference year

In the manuscript, the citation “Armstrong et al., 2028” appears, which is likely a typographical error. Kindly ensure this is corrected throughout the document, including in figure legends, references, and any other relevant sections.

Our response: We have made the corrections. It should be Armstrong et al., 2018.

*4. I’ve been asked to comment on the analysis performed in response to Reviewer #1’s Point 3: I have a fundamental issue with the central assumption of the analysis. Specifically, I’m not sure that defining “dysbiosis” as a random Cauchy variable is correct. Typically, dysbiosis is computed from ecological distances or log-ratio transforms, not by assuming a particular parametric distribution for the underlying abundances or the index itself. Additionally, dysbiosis is an ecological term defined for the entire community, so it doesn’t make sense to me to state that “particular species were dysbiotic,” as suggested with the *Prevotella* example in the rebuttal.*

Our response: We agree that the main concern lies in the choice of terminology. Traditionally, dysbiosis has been defined at the sample level. Our intent was to innovate it by broadening the concept to the species level in order to gain deeper insights.

As noted in our response to Reviewer 1, and in the manuscript, prior studies have described dysbiosis of samples in terms of (i) alpha diversity (Le Chatelier et al., *Nature*, 2013), (ii) differential abundance of taxa between groups (Gevers et al., *Cell Host & Microbes*, 2014), or (iii) beta diversity (Lloyd-Price et al., *Nature*, 2019). In contrast, our approach goes a step deeper by quantifying disruption at the individual taxon level.

Having said that, to avoid any confusion, we replaced the term “dysbiosis” with “disruption.” Accordingly, our measure would now be called the **taxon-specific measure of DISruption in COrrrelations (DISCO)**. In essence, DISCO quantifies how disrupted the correlations are for a given taxon in a specific environment relative to a reference environment. As suggested by the editor, we have made changes to the Abstract as well to reflect this terminology.

As demonstrated in our HIV dataset, as well as across the four external datasets suggested by the editor, and simulated data (as per reviewer 1’s suggestion), DISCO is useful in identifying taxa with disrupted correlations.

For clarity, as a minor point, we note that our method does not assume any parametric model, but it rather leverages the theoretical result that the proposed statistic is Cauchy distributed (Liu & Xie, *JASA*, 2020). The Cauchy transformed statistic of p-values is widely applied in high-dimensional genetics and genomics analyses. Like a Z-score, it is unit-free, but instead of the normal distribution, it is based on the Cauchy distribution.

Reviewer #2 (Remarks on code availability):

The results presented in the paper are reproducible, and the code serves as a useful resource for the community. The code includes a README file with instructions for installation and

execution. It is possible to install and run the code.

Our response: Thank you.